

# Global analysis for periodic variations of gravity wave squared amplitudes and momentum fluxes in the middle atmosphere

Dan Chen[1,2], Cornelia Strube[2], Manfred Ern[2], Peter Preusse[2], and Martin Riese[2]

[1]Key Laboratory of Meteorological Disaster, Ministry of Education/Joint International Research Laboratory of Climate and Environment Change/Collaborative Innovation Center on Forecast and Evaluation of Meteorological Disasters, Nanjing University of Information Science & Technology, Nanjing, China
[2]Institute of Energy and Climate Research - Stratosphere (IEK-7), Forschungszentrum Juelich GmbH, Juelich, Germany

**Correspondence:** Dan Chen (nuistcd@163.com)

**Abstract.** Atmospheric gravity waves (GWs) are an important coupling mechanism in the middle atmosphere. For instance, they provide a large part of the driving of long-period atmospheric oscillations such as the quasi-biennial oscillation (QBO) and the semiannual oscillation (SAO) and are in turn modulated. They also induce the wind reversal in the mesosphere - lower thermosphere region (MLT) and the residual mean circulation at these altitudes. In this study, the variations of monthly

zonal mean gravity wave square temperature amplitudes (GWSTA) and, for a first time, absolute gravity wave momentum flux (GWMF) on different time scales such as the annual, semiannual, terannual and quasi-biennial variations are investigated by spectrally analyzing SABER observations from 2002 to 2015. Latitude-altitude cross sections of spectral amplitudes and phases of GWSTA and absolute GWMF in stratosphere and mesosphere are presented and physically interpreted. It is shown that the time series of GWSTA/GWMF at a certain altitude and latitude results from the complex interplay of GW sources,

propagation through and filtering in lower altitudes, oblique propagation superposing GWs from different source locations and, finally, the modulation of the GW spectrum by the winds at a considered altitude and latitude. The strongest component is the annual variation, dominated on the summer hemisphere by subtropical convective sources, and on the winter hemisphere by polar vortex dynamics. At heights of the wind reversal also a 180° phase shift occurs, which is at different altitudes for GWSTA and GWMF. In the intermediate latitudes a semi-annual variation (SAV) is found. Dedicated GW modeling is used to

investigate the nature of this SAV, which is a different phenomenon from the tropical SAO also seen in the data. In the tropics a stratospheric and a mesospheric QBO are found, which are, as expected, in anti-phase. Indication for a QBO influence is also found at higher latitudes. In previous studies a terannual variation (TAV) was identified. In the current study we explain its origin. In particular the observed patterns for the shorter periods, SAV and TAV, can only be explained by poleward propagation of GWs from the lower stratosphere subtropics into the mid and high latitude mesosphere. In this way, critical wind filtering in

the lowermost stratosphere is avoided and this oblique propagation hence is likely an important factor for MLT dynamics.

## 1 Introduction

Gravity waves (GWs) are oscillations in the atmosphere that result from a disturbance in the general atmospheric flow and are balanced by an interplay of gravity and buoyancy. Their most prominent sources are orography, convection, fronts and



spontaneous adjustment, which are mostly located at low altitudes in the troposphere. As GWs propagate upwards, they carry GW momentum flux (GWMF) and, by interaction with the background flow, accelerate or decelerate the mean winds. This facilitates a coupling of the different layers of the atmosphere. Gravity waves often break while propagating upward, either to compensate for the amplitude growth due to exponential density decrease or because of critical level filtering. The dissipation

of GWs results in deposition of momentum and energy and, hence, accelerate or decelerate the atmospheric flow (Lindzen, 1981; Holton, 1983). Additional processes that may lead to GW dissipation are radiative and turbulent damping (e.g. Marks and Eckermann, 1995). Even without dissipation GWs may exchange momentum and energy with the background wind, either by horizontal refraction (Buehler and McIntyre, 2003; Preusse et al., 2009) or by transient non-linear interactions (Sutherland, 2006; Muraschko et al., 2015; Boeloeni et al., 2016).

Gravity waves strongly influence the global circulation and the temperature structure in the middle atmosphere (Fritts and Alexander, 2003). Several observed major phenomena cannot be explained by radiative balance or large-scale waves, like Rossby waves, alone; sometimes the structure expected from radiative balance is even reversed. Examples for this are the warm winter polar stratopause, the cold summer mesopause and the wind reversal at the mesopause (e.g. Hitchman et al., 1989; McLandress, 1998; Tomikawa et al., 2008; Siskind, 2014). Gravity wave driving of the atmospheric circulation gives

an explanation for these phenomena. Furthermore, both the Quasi-biennial oscillation (QBO) and the Semiannual oscillation (SAO) are, in large part, driven by atmospheric GWs (e.g. Dunkerton, 1997; Alexander and Ortland, 2010; Ern et al., 2014, 2015). Accordingly, GWs and their effect on the mean flow have to be included in general circulation models in order to reproduce the general structure of the atmosphere in a realistic way (Kim et al., 2003; Alexander et al., 2010). However, current-day implementations of GWs still fail to correctly generate some major zonal mean circulation patterns, as e.g. the

study by Garcia et al. (2017) on the southern hemisphere winter polar vortex shows.

Only a few high-resolution general circulation models (GCMs) are able to explicitly resolve a larger part of the whole spectrum of GWs and to simulate a reasonably realistic middle atmosphere without the use of a GW parametrization (e.g. Watanabe et al., 2008). Traditionally, a dedicated GW parametrization scheme is used for orographic waves; GWs from all other sources are accounted for in a so-called non-orographic GW parametrization (e.g. Hines, 1997; Alexander and Dunkerton,

1999; Warner and McIntyre, 2001). Attempts are made recently to replace the unspecific non-orographic GW parametrization schemes by parametrizations tied to physical source processes (e.g. Chun and Baik, 1998; Song et al., 2007; Richter et al., 2010; Kim et al., 2013; Plougonven et al., 2017). Obviously, only by using GW parametrizations tied to physical source processes, a full feedback towards weather and climate changes can be realized. Introducing GW parametrizations based on source-processes, e.g. dedicated convective parametrizations (Chun and Baik, 1998; Beres et al., 2005), introduces new tunable

parameters in the models which have to be constrained by observations (Choi et al., 2009, 2012; Trinh et al., 2016). Additional challenges are the stochastic nature of GWs (Hertzog et al., 2012; de la Camara et al., 2014) and, in particular, the oblique propagation of GWs (Sato et al., 2003; Preusse et al., 2009; Sato et al., 2009; Kalisch et al., 2014; Ribstein et al., 2015). While in conventional column-based GW parametrizations a GW acts at the geographic location of its source, lateral propagation may distribute GW momentum to different regions. As a recent case study shows (Krisch et al., 2017), this distribution may

take place already in the lowermost stratosphere. Lateral propagation also raises the question whether the attribution of GWMF

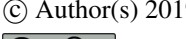



to sources by spatial correlations is adequate (Preusse et al., 2014). All this makes the validation of models with observations extremely important but also challenging.

A comparison between GWMF in observations and climate models has been carried out by Geller et al. (2013). Distributions of GWMF from two high-resolution GCMs explicitly resolving GWs and three GCMs employing GW parametrizations are

compared to three different satellite data sets and observations by super-pressure balloons. The study considers distributions for both northern and southern winter and northern and southern summer. This gives a first valuable overview of global GW distributions, but only provides little hints to the active source processes and GW phase speeds. Though the comparison by Geller et al. (2013) shows good agreement in relative distributions of spatial patterns among the models and the observations, they are inconsistent in details. For example, at summer high-latitudes most models have larger GWMF than indicated by

the observations. In addition, the satellite-derived GWMF has stronger vertical gradients than the model fluxes, and the high-resolution models are resolving the momentum fluxes of important orographic GW sources not well. Still many of these points remain inconclusive because of the large errors associated with the GWMF observation (Ern et al., 2004, 2018). Temporal variations are more sensitive to influences of source and interaction processes and thus can provide more detailed constraints for global models. It has been shown, by Preusse et al. (2009) for instance, that considering the seasonal cycle may give

valuable hints on sources as well as on phase speed distributions.

There are a number of previous studies revealing variations of different periods from long-term GW observations. Some of the most relevant are discussed in the following paragraph. For instance, GW activity exhibits a prominent annual cycle over mid-latitudes in the stratosphere (Li et al., 2010; John and Kumar, 2012), and a semiannual signal is found in the mesosphere, which is caused by the seasonal change of mesospheric winds (Yang et al., 2010). From the upper troposphere to the lower

stratosphere in the tropics, GWs exhibit variations related to the QBO (Vincent and Alexander, 2000; John and Kumar, 2012; Ern et al., 2014), while a semiannual oscillation is found in the tropical stratosphere (Zhang et al., 2012), stratopause region (Ern et al., 2015), and the mesosphere (Antonita et al., 2008). In addition, Krebsbach and Preusse (2007) reported on a four-monthly (terannual) variation of GWs, and Shuai et al. (2014) showed that it appears at mid-latitudes between 85 and 115 km. Recently, some studies have inferred the 11-year solar cycle of GW activity in the tropics from observations (Ern et al., 2011;

Li et al., 2016; Liu et al., 2017). Ern et al. (2011) and Liu et al. (2017) showed that there is an anti-correlation between the 10.7 cm solar flux and GW amplitudes respectively GW potential energy. Li et al. (2016) further find indication that this variation of GW activity may be linked to variations in convective activity. Long-term observations of GW variances and GWMF are available from a number of ground-based observations (e.g. Tsuda et al., 1990; Espy et al., 2004; Hoffmann et al., 2010), but from these it is difficult to compare the intensity of inter-annual and intra-annual variability at different latitudes. However, a

full global picture of the temporal spectrum of GW variations is still missing. This is in particular the case for GWMF, though it is a parameter that is directly linked to the potential of GWs to drive the global circulation. The global picture gained from satellite observations facilitates understanding of the physical mechanisms that couple different altitudes and latitudes.

Based on 13 years of data from the Sounding of the Atmosphere using Broadband Emission Radiometry (SABER) instrument, we focus on analyzing temporal variations of square-root zonal average GW square temperature amplitudes (GWSTA)

and GWMF variations in the latitude range from 50°S to 50°N continuously covered by SABER. Four different frequencies,





corresponding to terannual up to quasi-biennial periods, will be investigated. The variations will be interpreted in terms of different source mechanisms and propagation pathways. In addition, we utilize the length of the data set to look for decadal variation features. Eventually, we will show that the distributions can be consistently extended also to latitudes of 80° on both hemispheres using sinusoidal fits, which allows to disregard the gaps in the temporal coverage caused by SABER yaw

maneuvers.

The remainder of this paper is organized as follows. The data of GWSTA and absolute GWMF that are obtained from SABER temperature observations, as well as a data set based on combining European Centre for Medium-range Weather Forecasts (ECMWF) analysis data with GROGRAT ray-tracing are described in Sect. 2. In Sect. 2 also the used spectral analysis methods is introduced. In Sect. 3, annual, semiannual, terannual and quasi-biennial variations of GWSTA and absolute

GWMF in the latitude range of 50°S to 50°N and altitude range from 30 to 90 km are analyzed in detail. The 11-year solar cycle is evaluated using average GWSTA and GWMF over the latitude range of 25°S to 25°N. In Sect. 4, an extension of the analysis towards high-latitudes is presented. A systematic summary and conclusions are given in Sect. 5.

## 2   Data and methods

### 2.1   GWSTA and GWMF data from the SABER instrument

Our study is based on temperature data observed by the infrared limb sounding instrument SABER which was launched on board of the TIMED (Thermosphere Ionosphere Mesosphere Energetics and Dynamics) satellite in December 2001. The SABER instrument started its observations in January 2002 and was designed to measure in the altitude range from near the tropopause to the lower thermosphere ($\sim$20 km to 120 km). The temperatures are derived from $15\mu m$ $CO_2$ infrared emissions. Details about the temperature retrieval are given in Remsberg et al. (2008). SABER observes the atmosphere in limb geometry

and has a 2 km vertical field of view. This makes the instrument sensitive to atmospheric GWs with horizontal wavelength longer than $\sim$100 km and vertical wavelengths longer than $\sim$4 km (Preusse et al., 2002).

In this study, data from February 2002 to January 2015 are used. This continuous 13-year data set allows for studying the systematic intra-annual, annual, and inter-annual structures of the global distribution of GWs. A detailed description of the GWSTA and absolute GWMF data used in this study is given by Ern et al. (2011) and Ern et al. (2018). We summarize briefly

how the monthly mean zonal averages are calculated in the following paragraph.

In order to obtain altitude profiles of temperature fluctuations that can be attributed to small-scale GWs, the atmospheric background temperature consisting of zonal average and large-scale waves (i.e. planetary waves and tides) is subtracted from each altitude profile of SABER temperatures; for details see Ern et al. (2011) and Ern et al. (2013). It should be noted that this background estimate is also capable to remove large scale inertial instabilities. In this way biases are avoided that can strongly

affect GW estimates obtained from a merely vertical profile analysis (Rapp et al., 2018). In a next step, we derive for each altitude profile of SABER temperature fluctuations, GW temperature amplitudes, vertical wavelengths and phases for sliding vertical windows of 10 km vertical extent (Preusse et al., 2002). Based on the method of Ern et al. (2004, 2011), absolute GWMF is derived from vertical phase differences for those consecutive pairs of altitude profiles in the SABER measurement

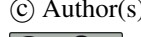



track that have short enough along-track distances (shorter than $\sim$300 km) and that at the same time agree in their GW vertical
wavelengths (differences should not exceed 40%). Using these criteria, at good likelihood the same GW is observed in both
altitude profiles of a pair. In addition, undersampling of the horizontal wave structure of the observed GWs is minimized. Still,
errors of absolute GWMF are at least a factor of two (Ern et al., 2004).

The calculation of monthly zonal average GWSTA and absolute GWMF is carried out as follows: First, global distributions
of monthly averaged GWSTA and GWMF are calculated on a regular grid in longitude and latitude for a set of given altitudes.
The grid resolution is $10°$ in longitude and $5°$ in latitude. To obtain values at each grid point, the data are averaged in grid boxes
of $30°$ longitude times $20°$ latitude centered at each grid point (the grid boxes are overlapping in both longitude and latitude
direction). Finally, to obtain zonal averages, the gridded data is averaged along the longitude direction. The effective latitude
resolution of the resulting zonal averages is $20°$, corresponding to the latitude extent of the grid boxes used. The data set is
available at https://doi.org/10.1594/PANGAEA.879658, and a full description of this data set is given in Ern et al. (2018).

In summary, the monthly mean values entering the time series analysis for GWSTA are calculated as follows:

$$\bar{\Gamma} = \sqrt{\langle \hat{T}^2 \rangle}, \tag{1}$$

where $\langle\ \rangle$ denotes the average as described above, $\hat{T}$ is the temperature amplitude. The values for GWMF are calculated
according to

$$\bar{F_{ph}} = \langle \frac{1}{2}\bar{\rho}\left(\frac{g}{N}\right)^2 \frac{k_h}{m}\left(\frac{\hat{T}}{\bar{T}}\right)^2 \rangle \tag{2}$$

with $\bar{\rho}$ the background density, $\bar{T}$ the background temperature, $g$ gravity acceleration and $N$ the buoyancy frequency. Also,
the horizontal wavenumber $k_h$ and the vertical wavenumber $m$ enter the momentum flux. The background atmosphere, and
in particular $\bar{\rho}$ and $\bar{T}$, vary with season. Compared to GWSTA this induces additional variations into GWMF and hence can
modify the altitude-latitude pattern of amplitudes and phases of temporal variations such as annual and semiannual variation.

## 2.2   Zonal and absolute GWMF from a combination of ECMWF analysis data and GROGRAT simulations

The European Centre for Medium Range Weather Forecast (ECMWF) provides analysis fields from their Integrated Forecast
System (IFS) run in operational numerical weather prediction. These global atmospheric data have high spatial resolution
and allow to infer information about GWs including GWMF and propagation direction of the waves. We use data from the
years 2014 and 2015 to investigate how the loss of direction information when calculating GWMF and GWSTA influences
the attribution of the variations to semiannual and annual signal. In these years, the ECMWF IFS ran with resolution settings
(T1279, N640) corresponding to a horizontal grid resolution of approx. 16 km. Using the method of Skamarock (2004),
Preusse et al. (2014) showed that the shortest horizontal wavelengths resolved adequately with this resolution are of the order





of 200 km. This scale is of similar magnitude as the shortest scales visible in limb sounding observations (Preusse et al., 2002). Furthermore, several studies have shown that GWs resolved by the ECMWF IFS match observations in the lower stratosphere well (e.g. Schroeder et al., 2009; Shutts and Vosper, 2011; Plougonven et al., 2013). At higher altitudes, the vertical resolution decreases and a sponge layer is employed. Accordingly, GW amplitudes and GW momentum flux spuriously decrease above

40 km. In order to understand GWs in the entire altitude range of the stratosphere and mesosphere, we analyze GWs in the ECMWF IFS temperature data at 25 km and propagate these waves upward with the Gravity wave Regional Or Global RAy Tracer (GROGRAT). Thus using GROGRAT, we can reconstruct a consistent picture of GW variance and momentum flux from 25 km to the mesopause. In the following, we briefly describe the wave analysis technique, the ray-tracer, the construction of the background atmosphere and the evaluation of the generated ray paths.

**2.2.1   Wave analysis technique**

In order to analyze the ECMWF IFS data with respect to GWs, we use the small-volume three-dimensional sinusoidal fit technique, called S3D, described in Lehmann et al. (2012). The data set is partitioned into non-overlapping cuboids of $1.8°$ latitude $\times$ $1.8°$ longitude $\times$ 10 km altitude with a center altitude of 25 km. In these cuboids, the three-dimensional wave vector, amplitude and phase of the sinusoid describing most of the variance are determined using a least-square fit. By these parameters

the fitted wave is fully characterized and rays can be launched from the center point of the respective cuboid (Preusse et al., 2014; Krisch et al., 2017).

**2.2.2   Ray-tracing with GROGRAT**

The employed ray-tracer GROGRAT (Marks and Eckermann, 1995) is based on the WKB assumption and derives the group velocity of a GW from the GW dispersion relation. This allows to estimate the location of a GW packet after a time step

according to the ray-tracing equations (Lighthill, 1978; Marks and Eckermann, 1995). The new location is associated with a different background state in terms of different winds and buoyancy frequency. This induces changes of the wave vector. Together, location vector and wave vector form the state vector of the GW packet which is projected in time using a fourth-order Runge-Kutta scheme. The GW amplitude and momentum flux are calculated based on the conservation of wave action flux along the ray. GROGRAT takes GW saturation due to static and dynamic instability into account as well as GW damping

due to radiative and turbulent dissipation.

The background atmosphere is defined by the large-scale horizontal wind and buoyancy frequency fields, which the ray-tracer uses to calculate the GW propagation. In order to construct an appropriate background atmosphere, we use a combination of ECMWF IFS data up to 45 km and temperature observations as well as geostrophic wind fields from SABER above 35 km. A smooth transition between the two fields is applied for the altitude range from 35 km to 45 km altitude. In addition, a profile-

wise bias correction based on the respective average values of the two fields in this altitude range is applied to the SABER temperatures. In this way, large temperature gradients and enhanced or reduced values of the buoyancy frequency are avoided. The SABER geostrophic winds reach up to ~100 km and contain the zonal mean and planetary wave structures. Tides are not explicitly taken into account. The latitude coverage changes with yaw maneuver and is the same as for the GWMF data.





### 2.2.3 Absolute and zonal GWMF from ray-traces

The ray-tracer is run based on daily ECMWF data for 12 UTC for the years 2014 and 2015. The individual ray-traces are interpolated to fixed altitudes between 25 km and 90 km in 5 km steps. Data are then binned into 3.6° latitude bins, the extent of two S3D cuboids. In order to determine the average GWMF at one altitude level, we normalize the calculated GWMF with

the number of propagating rays at the launch altitude. In this way, total GWMF of all ray-traces decreases with altitude due to dissipation and critical level filtering of the individual GWs. In some regions, rays may converge due to lateral propagation, in which case GWMF for this particular region also can increase. For comparison with the SABER data, monthly mean zonal averages of GWSTA and GWMF are generated.

## 2.3 Spectral analysis methods

We perform spectral analysis to identify the dominant temporal variations of GW activity. Monthly zonal means of GWSTA and absolute GWMF are calculated and then combined into a time series covering the 13 years considered. For better comparison with Krebsbach and Preusse (2007) we take the square-root of GWSTA for further analysis. The time series are analyzed using Fast Fourier Transform (FFT). For a cross check on the main frequencies identified with the FFT, Maximum Entropy Method (MEM) (Press et al., 2007) spectra are calculated in addition.

As an example, the results of FFT and MEM applied on absolute GWMF from the SABER data set are presented in Fig. 1. Figure 1a shows the spectral amplitudes of the FFT performed for 45°N and 50 km, Fig. 1b shows all FFT spectra in the latitude band from 50°S to 50°N for 30 km altitude, and Fig. 1c gives the corresponding MEM power spectral density results.

The FFT spectrum in Fig. 1a exhibits major frequencies at 2.167 years, 1.0 year and the harmonics of 0.5 and 0.333 years. In order to highlight significant spectral components, the dashed line indicates 1.3 times the average spectral amplitude. This

value is motivated as follows: If the spectrum would consist only of Gaussian white noise, 95 % of all spectral amplitudes would be below this value. This can be confirmed with a Monte Carlo experiment. If real spectral information is contained, this is a conservative estimate for a white noise floor. However, if there are variations longer than the largest period resolved, this will introduce a low-frequency background. The dashed line hence gives an orientation about which periods are likely to be significant. Further corroboration for dominant periods can be taken from the occurrence of the same periods in statistically

independent regions of the atmosphere.

The latitude-period cross section in Fig. 1b covers latitudes in the range from 50°S to 50°N. Due to solar angle restrictions of the TIMED satellite, only this latitude range is continuously covered by SABER observations and can be analyzed reliably by FFT and MEM, which require continuous and equidistant sampling. The color code shows the spectral amplitude. The black dots in the FFT cross section mark values below the significance level as defined above. In Fig. 1b the same periods as in

the 45°N example spectrum of Fig. 1a are prominent. In particular for values below 3 years, the significant periods are found over a wide range of latitudes. In addition, there is a quasi-biennial maximum at the equator. The 0.333 year period is most pronounced in the extra-tropics.



As an independent test, the same distribution but generated using the MEM analysis is shown in Fig. 1c. The prominent features from the FFT spectrum are reproduced by this spectral method.

In addition to the methods presented here, we use sinusoidal fits of the main spectral periods identified with FFT and MEM. This allows to extent the latitude coverage. The results of the sinusoidal fits are validated by comparison to FFT in the latitude
range 50°S to 50°N where both methods can be applied (see also Sect. 4).

## 3    Morphology of GWSTA and absolute GWMF variations on different time scales

In this section, we focus on the spatial distribution of the dominant spectral periods of 0.333, 0.5, 1.0 and 2.167 years identified in Fig. 1, which correspond to terannual, semiannual, annual, and quasi-biennial variations. In the following, we will discuss altitude-latitude cross sections for the prevailing variations of GW activity that are shown in Figs. 2, 3, 5, and 7, respectively.
The cross sections cover the height range between 30 km and 90 km within the latitude band 50°S to 50°N. The plots for the square-root of GWSTA are given in the upper row. The plots for absolute GWMF are given in the lower row. Values which fall below the significance level are indicated by contours (in white for GWSTA and in black for absolute GWMF) with ticks pointing into the insignificant area. Each of the figures shows the spectral amplitudes in the left column and the corresponding phase in the right column. For easy comparison with the fit results in Section 4, both the ranges of colorbar of
spectral amplitudes are the same in each investigated period. The shown phases belong to the maximum spectral amplitude and are inferred in months with respect to 1 January 2002. For better readability in particular of annual and semiannual variation, we start the colorbar in April. In the case of the annual variation for instance, we thus avoid a jump from red to blue between December and January (maximum for NH winter). In this case, red colors indicate that the phase of the maximum is around July, while blue and green indicate that the phase of the maximum is around January. However, the range of the colorbar
changes according to the investigated period, such that the same colors point to the same relative phase of that period, not necessarily the same month or season. In particular, the same color may mean two (semiannual) or three (terannual) different months indicating the same phase.

### 3.1    Annual variation (AV)

The annual variation is shown in Fig. 2. Spectral amplitudes of GWSTA exhibit a general increase with altitude and form at
most latitudes a local maximum around 65 km. Considering the latitudinal dependence of the spectral amplitude of GWSTA (Fig. 2a) for a given altitude below ∼70 km, we find four maxima: two stronger maxima at high latitudes and two weaker maxima at subtropical latitudes. The high-latitude maximum values are larger in magnitude in the Southern Hemisphere than in the Northern Hemisphere. The corresponding phases (Fig. 2b) of these two regions indicate that the maximum occurs in winter of the respective hemisphere, i.e., around June in the Southern Hemisphere and around December in the Northern
Hemisphere. This phenomenon is consistent with the recent studies on SABER GW distributions (e.g., Liu et al., 2017) and observations from different instruments (e.g., Alexander et al., 2008; Ern et al., 2011). Enhancements of GWSTA amplitudes at latitudes poleward of 40° are correlated with high westerly wind speeds that are associated with the winter polar vortices (cf.





also Preusse et al. (2004)). A likely reason for this correlation is that the strong background winds provide good propagation conditions and allow for high saturation amplitudes of upward propagating GWs (Preusse et al., 2006).

The secondary, much weaker local maximum that is found in the subtropics has maximum GWSTA values in summer of the respective hemisphere, i.e., December in the Southern Hemisphere and July in the Northern Hemisphere. This maximum of
GWSTA is likely related to convective GWs excited in the subtropics (e.g., Trinh et al., 2016).

Different from GWSTA, the spectral amplitude maxima of absolute GWMF (Fig. 2c) occur at lower altitudes and gradually decrease with altitude. The reason for this decrease is likely that GW dissipation takes place in the whole altitude range while GWs propagate upward. For each hemisphere, the latitudinal structure displays two similarly strong local maxima of the AV spectral amplitudes of GWMF, one peak at higher latitudes and a second peak in the subtropics. These are clearly visible at
altitudes of 30 to 70 km. Together with the associated phase information, this suggests that the annual variations are related to the source of the GWs as well as to the background winds, which is in agreement with the results for GWSTA that were discussed before. At high-latitudes, the maximum in the Southern Hemisphere has larger values in GW momentum flux than the maximum in the Northern Hemisphere. The corresponding phase at southern high-latitudes continuously descends from May in the mesosphere (about 70 km) to August in the lower stratosphere. Different from this, the phase at northern high-latitudes
has no strong altitude dependence below 70 km and is about constant around December (Fig. 2d). Because GW activity at high-latitudes is usually related to the winter polar vortex, a likely reason for the observed difference between Northern Hemisphere and Southern Hemisphere are structural differences of the winter polar vortex. The southern polar vortex is larger, stronger and more durable than the northern polar vortex. Further, the breakdown of the southern polar vortex happens gradually from high to low altitudes. This will gradually shift the amplitude maximum in the Southern Hemisphere to later in the year at lower
altitudes. Different from this, the breakdown of the northern polar vortex happens quite rapidly, which is related to the stronger activity of planetary waves in the Northern Hemisphere. Breaking of planetary waves can quite rapidly decelerate the polar vortex and lead to stratospheric warmings. A rapid breakdown of the polar vortex and the corresponding reduced wind speeds will lead to strong and sudden drops in GW activity over a large altitude range (e.g., Ern et al., 2016). Stratospheric warmings almost never occur in the southern hemisphere. Consequently, a slow phase shift of the altitude maximum in GWSTA and
absolute GWMF would be expected for the Southern Hemisphere but not for the Northern Hemisphere, which is in agreement with Fig. 2b and 2d.

In the subtropics, the area of enhanced amplitudes of absolute GWMF in each hemisphere spreads from the equator to 30°S with descending phase (November to January) and to 30°N with a quasi constant phase (June), respectively. As mentioned before, these maxima in the subtropical summer hemisphere are related to convectively excited GWs.
One important feature in Fig. 2c is that the amplitude peak of absolute GWMF in the subtropical summer hemisphere obliquely shifts poleward at increasing altitude up to the mesopause. This shift feature is in agreement with previous observational results by Ern et al. (2011), who found that GWs propagated poleward by about 15° between 30 and 70 km in January and July.

During the latitude shift, the longitudinal structure of GW activity is maintained (Ern et al., 2013). Also our phase results
shown in Fig. 2d are consistent with these findings. In the regions from 30° to 50° in both hemispheres above 70 km the phase





of the maximum spectral amplitude of GWSTA and absolute GWMF is not in winter as for lower altitudes, but in summer. That is, in the Southern Hemisphere, the phase of the upper-mesosphere maximum corresponds to November, and the phase gradually descends towards January in the subtropical stratosphere. In the Northern Hemisphere, the phase corresponds to a maximum in June in the upper mesosphere and descends towards July in the stratosphere. This means that the maximum

GW activity at latitudes between 30° and 50° occurs above 70 km in the same season as the maximum of GW activity in the subtropical stratosphere. These features lead to a phase jump at latitudes between 30° and 50° for both hemispheres around 70 km: the peak-time of the annual variation of GWSTA and absolute GWMF is during the winter months below 70 km and changes to summer months above 70 km. This is consistent with the results of Liu et al. (2017), who found the same characteristics for the annual peak-time of GW potential energy based on a 14-year SABER data set. The poleward shift of

upward propagating GWs suggests that GWs generated by convection likely contribute to the poleward tilt of the summertime mesospheric jets and the wind reversal in the summer upper mesosphere at mid- to high-latitudes.

However, comparing the details of Figs. 2b and 2d, we find a difference between GWSTA and absolute GWMF in the altitude of the phase reversal. In absolute GWMF, the phase jump occurs at about 70 km in both hemispheres, while in GWSTA it occurs around 80 km, i.e., there is a difference of 10 km. One possible reason for this effect could be the variation

of background density. It is known that for a fixed altitude there is a seasonal variation in density. There is higher pressure at the pole during summer and lower pressure during winter. In our study we use the parameter absolute GWMF in units of $\mathrm{Pa}$, which includes the background density. As a cross check, we tested the parameter absolute GWMF in units of $\frac{\mathrm{m}^2}{\mathrm{s}^2}$, which does not contain the background density. The results show that the effect of background density explains about 50% of the altitude difference ($\sim$5 km) between the phase shift in GWSTA and absolute GWMF.

**3.2   Semiannual variation (SAV)**

In this section we will consider the distribution of semiannual variations (SAV) of spectral amplitudes of GWSTA and absolute GW momentum flux. We intentionally do not use the term semiannual oscillation (SAO), since this is conventionally used for the wave-driven oscillation in the tropical mesosphere. Whether and where the SAV patterns analyzed here are connected with the SAO will be discussed.

The most prominent feature of the SAV in GWSTA (see Fig. 3, upper row) are two areas of enhanced spectral amplitudes with a strong poleward tilt. They extend from about 20° latitude at 60 km altitude to 50° latitude at 90 km altitude for the respective hemisphere. This variation is stronger in the Southern Hemisphere than in the Northern Hemisphere, similar to the annual variation.

Together these two maxima form a crescent-shaped region of enhanced amplitudes, decreasing in magnitude towards the

equator. The distribution and shape of the SAV spectral amplitudes for GWSTA is consistent with previous findings by Krebsbach and Preusse (2007), who used a 4-year SABER data set. In almost all regions, except for the tropical mesopause, the phases correspond to the maximum GWSTA taking place in June/July or December/January. Considering SAV and AV patterns together, this means that there is a semiannual variation with a larger peak in the summer months and another peak in the winter months.



For the SAV of absolute GWMF, the subtropical maxima (Fig. 3c) shift polewards at higher altitudes, similar as observed for the annual variation. The phase distributions (Fig. 3d) are similar to those of GWSTA.

Combining all this information suggests that the crescent-shape of enhanced GWSTA spectral amplitudes for the SAV is caused mainly by convective GWs from the subtropical summer hemisphere that propagate upwards and obliquely to higher

latitudes in summer in combination with GWs related to the polar vortex in winter, thus forming two maxima per year. Where these two maxima overlap they result in an SAV. This is why the SAV appears at mid- to high-latitudes (30° to 50°) in the mesosphere. This structure was also found in previous studies, for example, Fig. 9e in Preusse et al. (2009) and Fig. 7c in Ern et al. (2011).

### 3.2.1  Is the semiannual actually an annual variation?

However, it is known that GWs preferentially propagate eastward in summer (against the easterly background winds) and westward in winter (against the westerly background winds) (e.g., Ern et al., 2017). Theoretically, this should cause an annual rather than a semiannual variation for GW momentum flux if we could take the propagation direction into account. Hence, the apparent SAV of absolute momentum flux at mid- to high-latitudes in the mesosphere can be expected to be an annual variation if the direction of GWMF is considered. We further investigate this hypothesis by performing spectral analysis on zonal GWMF

and absolute GWMF based on GROGRAT model simulations (setup as described in Sect. 2). The result is illustrated in Fig. 4. Figure 4a and 4d shows the semiannual and annual spectral amplitude of absolute GWMF from GROGRAT, respectively. Similarly, Fig. 4b and 4e show the semiannual and annual spectral amplitudes of the zonal component of GWMF, which can be either eastward/positive or westward/negative depending on the preferential propagation direction. In addition, Fig. 4c and 4f show the difference between spectral amplitudes of absolute GWMF and zonal GWMF for the annual and semiannual

variations, respectively. There is a qualitative likeness of the structures in Fig. 4a with Fig. 3c and Fig. 4d with Fig. 2c indicating that the important processes are captured in the simulation. We therefore use the model results to qualitatively address, how various features of the seasonal cycle of GWMF enter the spectral distributions deduced from the observations.

In order to understand the results, let us first perform a thought experiment with different idealized seasonal cycles. At first assume an AV of a pure sinusoid with equally large positive and negative zonal GW momentum flux values. In the spectral

analysis this would result in an AV only. The seasonal cycle formed by the absolute values has two positive maxima. In the spectral analysis this would result in a SAV and its harmonics, but not in the AV. Now assume a single positive maximum. This would result in a strong AV and decreasing signals for the harmonics, depending on the shape of the peak. This will be the same for the absolute values. Likewise a single minimum (negative maximum) will enter the spectral analysis results of direction-resolved and absolute values the same way. If we now calculate for both AV and SAV the difference of the spectral

amplitudes for absolute values minus those for direction resolved values, we have the following expectations: An AV formed by a single maximum (e.g. due to the winter polar vortex or convection) will not show up in the difference. An AV formed by a sign reversal will show as a negative difference for the AV and a corresponding positive difference for the SAV.

Figures 4c and 4f show such matching patterns of negative AV and positive SAV differences between absolute and direction-resolved GWMF structures. They are found at latitudes 35° to 40°S around 70 km altitude and at 40°N around 60 km altitude.



Accordingly, the most prominent SAV features in Fig 3 are not signs for effects of a semiannual oscillation at higher latitudes but AV features involving a reversal of the propagation direction.

### 3.2.2 Relationship between SAV and the SAO

There is one spectral amplitude maximum of both GWSTA and absolute GWMF which likely is connected to the SAO. This
is a feature of high spectral amplitudes centered in the tropical ($10°$S to $10°$N) Mesosphere-Lower Thermosphere (MLT) region between 70 and 80 km altitude. The associated phase of the maximum GW activity is August/September (respectively February/March), which is about one to two months delayed relative to the stratosphere at these latitudes.

Two effects may be responsible for the amplitude enhancement in the mesosphere: First, selective filtering of convective GWs by the SAO in the tropical stratosphere and stratopause region (e.g., Ern et al., 2015) contribute to the forcing of the
strong SAO of zonal winds in the mesosphere (Smith, 2012). In return, modulation of GWs by the SAO in zonal winds in the tropical mesosphere result in a SAV of GW activity (Hirota, 1978; Burrage et al., 1996). Thus, the SAV of GW activity in the mesosphere may be coupled with the variation and filtering of GWs in the stratosphere and stratopause region.

### 3.3 Terannual variation (TAV)

Next to the periods already discussed, Fig. 1 shows a 4 months variation strongly enhanced above the background periods, that
has a prominent peak at mid- to high-latitudes. This variation has not been reported often before. Krebsbach and Preusse (2007) noted that a significant peak of four month period existed in the spectrum of their 4-year SABER data set without discussing this feature in detail. Shuai et al. (2014) used a 9-year SABER data set to show that GW activity exhibits a four-monthly variation at about 100 km altitude in the latitude range $40°$ to $50°$.

The terannual variation in GWSTA mainly appears at mid- to high-latitudes in both hemispheres from the upper stratosphere
to the mesosphere with a peak around 70 km altitude (Fig. 5a). The associated phase indicates a maximum of GWSTA in the first of the four months period (i.e. January/May/September) in the Southern Hemisphere and in the third of the four months period (i.e. July/November/March) in the Northern hemisphere (Fig. 5b), and hence is compatible with a maximum in mid-summer, i.e. January and July, respectively. In absolute GWMF, the feature of a poleward shift at higher altitudes, which has already been discussed for the distribution of the annual and semiannual variations, is even more prominent for the terannual
variation (Fig. 5c). The associated phases are the same as for GWSTA.

In order to understand the cause of the terannual variation, the time series of monthly zonal mean absolute GWMF from SABER data at 30 km altitude and at $20°$ respectively for the southern and northern hemisphere are shown in Fig. 6. We now introduce the annual average of absolute GWMF as a threshold for each year (blue lines). The values that exceed this threshold can be considered to represent an "active period" of GW activity in the subtropics. Accordingly, values below this
average represent a relatively "calm period". We can see that the "active period" lasts for 4 months each year. In the Northern Hemisphere, the active period spans from June to September and in the Southern Hemisphere from December to March. This corresponds to a high occurrence rate of tropical convective systems, such as tropical cyclones and convection related to the summer monsoon. Spectral decomposition of such a combination of 4 months active and 8 months calm period results first in





an AV component, but with much stronger amplitudes of the harmonics and in particular for a terannual variation than would result from a broader maximum. Therefore, the terannual variation of GWSTA and absolute GWMF is likely related to the duration of "active" convection in the tropics.

It should be noted that the terannual variation of GWSTA and absolute GWMF in the Northern Hemisphere is slightly
stronger than in the Southern Hemisphere, which is different from the semiannual and annual variations. This implies that subtropical convection is a likely factor here, since GWs forced by convection in the subtropics are more active in the Northern Hemisphere due to the particular strength of the Asian monsoon.

In summary, this confirms that subtropical summer hemisphere GWs related to convection are involved, and the structures show that they propagate obliquely to higher latitudes while propagating upwards.

**3.4   Quasi-biennial Oscillation (QBO)**

The spectral amplitudes of GWSTA and absolute GWMF for the quasi-biennial variation are usually weaker than those of annual, semiannual and terannual variations, still, they exhibit important features. These features can be summarized by the following three points:

Firstly, there are enhanced amplitudes related to the well-known QBO of the zonal winds in the equatorial stratosphere.
This enhancement is found in the latitude band between 10°S and 20°N and in the altitude range of 30 to 45 km, as indicated by significant spectral amplitudes of GWSTA (see Fig. 7a). According to the dominant QBO period found by the FFT, we use 26 months for the phase display of the QBO and use the number index 1 to 26 for the phase colorbar. The associated phases of GWSTA are descending from month 9 to month 13 (from orange to yellow in Fig. 7b). The odd years 2003, 2005, 2007, 2009, 2011, etc., are consistent with the easterly phase of QBO winds (cf. Figs. 2 and 4 of Ern et al. (2014)). This
enhancement of spectral amplitudes, which is not obvious in the annual, semiannual or terannual variations, suggests an effect of the equatorial QBO winds on gravity wave activity. Also for absolute GWMF enhanced spectral amplitudes are found in the tropical stratosphere. However, amplitudes are significant only between 30 and 40 km altitude (Fig. 7c).

Secondly, enhanced spectral amplitudes of the QBO in GWSTA and absolute GWMF can also be found near 50° latitude in both hemispheres in the stratosphere (see the regions marked as significant in Fig. 7). The strongest GW activity (maximum
GWSTA and GWMF) there is associated to the winter polar vortex and GWMF in summer is orders of magnitudes smaller. The signal of the quasi-biennial variation at these latitudes hence can only be carried by the modulation of the winter peak. Hence, this quasi-biennial variation of GWs is likely connected to the winter polar vortex (Baldwin et al., 2001) and partly with Sudden Stratospheric Warming events that frequently happen in Northern Hemisphere winters (e.g., Wang and Alexander, 2009; Wright et al., 2010; Ern et al., 2016).

Thirdly, there is a strong maximum of GWSTA and absolute GWMF spectral amplitudes around the equatorial mesopause (70 to 85 km). For clarification, we hereafter call the QBO of GWs in the stratosphere "SQBO", and the QBO of GWs in the mesopause region "MQBO". In the equatorial mesosphere, the associated phases of maximum GWSTA and absolute GWMF are around month 22/23 (blue color in Fig. 7b) and month 24/25 (light blue color in Fig. 7d), respectively, which lags about one year behind the respective SQBO phases. This implies that the MQBO may be coupling with the SQBO. de Wit et al. (2013)



used zonal-wind measurements located at Ascension Island (8°S) to study the interannual variability in the mesosphere (∼ 78 to 100 km). They detected an MQBO of winds with a period of 27.5 months in their observations. Furthermore, they found that the phase of the zonal wind SQBO at 30 km and the phase of zonal wind MQBO are anti-correlated, that is the MQBO is out-of-phase with the SQBO. This corresponds to a phase difference of about 180°: if, for example, the SQBO is in its easterly

phase, the MQBO is in its westerly phase. This anti-phase relationship between zonal wind SQBO and MQBO is qualitatively in good agreement with the anti-phase relationship between the SQBO and MQBO in GWSTA and absolute GWMF found in our study: The phase of SQBO is around month 9 to 13 while the phase of MQBO is around month 22 to 25 of the second wintertime/spring.

The SQBO-MQBO related variation of GWs in the equatorial region is further investigated in Fig. 8. Figure 8 shows time

series of equatorial GWSTA (Fig. 8a) and absolute GWMF (Fig. 8b) at 80 km and 35 km altitude, respectively. Shown are the averages for the individual months as well as a 13 months running mean, which highlight the QBO signal. In the equatorial stratosphere, we find clear indication of enhanced GWSTA and absolute GWMF in the years 2003, 2005, 2007, 2009, and 2011, which coincides with the easterly phase of the QBO winds in the middle stratosphere (see also Fig. 4 in Ern et al. (2014)). Comparing the 13 months running mean for 80 km (red lines) with the corresponding 13 months running mean for 35 km

altitude (green lines), we clearly find an anti-correlation between the MQBO and the SQBO: when GW activity is high in the stratosphere, this corresponds to low GW activity in the mesosphere, and vice versa. This anti-correlation of GW activity is consistent with the out-of-phase relationship of QBO winds in stratosphere and mesosphere as found by de Wit et al. (2013). This suggests that the modulation of GWs by the MQBO winds can account for the MQBO in GW activity, and the MQBO may be coupled to the SQBO. Please note that in the paper by Krebsbach and Preusse (2007) no obvious MQBO signal is

found. One possible reason could be the study of Krebsbach and Preusse (2007) was based on only 4 years of data (January 2002 to January 2006), which may be not sufficient to accurately detect a signal with a period of about 26-28 months.

### 3.5   How much of the variance is described by the main components?

In the previous four subsections, we focused on discussing the structure of the amplitude and phase of the four strongest spectral components. The question remains, whether these four selected periods well describe the total variance in the data. In

order to address this question, we compare in Fig. 9 the total variance with the variance by the reconstruction from the four leading periods. The full variance calculated from the analyzed time series of monthly mean GWSTA and GWMF data from SABER are shown in Fig. 9a and 9b. From the spectral amplitudes and phases of the four leading periods (TAV, SAV, AV and QBO) we reconstruct these time series and calculate the variance based on only these four components (Fig. 9c and 9d). Finally, we normalize the variance due to the leading spectral components by the total variance and show the ratio in Fig. 9e

and 9f. These two panels hence indicate, how much of the total variance is described by the four leading components, ranging from 0 (completely undescribed) to 1 (all variance described by the four leading components).

In most regions values are above 0.8 indicating that all important variations are captured. Regions with values of 0.4 or below are the tropical lower stratosphere and tropical mid-mesosphere, which are dominated by the QBO. Apparently the GW signal due to the QBO is not sufficiently regular to be completely described by a constant period of 26 months. This is consistent with




previous investigations of a varying QBO period (e.g. Salby and Callaghan, 2000). A higher degree of described variance may be reached by EOF methods, however, which are more difficult to interpret (Dunkerton, 2000, 2017). Furthermore, described variance in the mid-latitude upper stratosphere is higher in the southern than in the northern hemisphere. This is likely an effect caused by sudden stratospheric warmings which terminate the northern winter vortex at different times, but do not occur on the southern hemisphere (with the exception of the split-vortex event in 2002).

## 3.6 Gravity wave variations potentially related to the 11-year solar cycle

The SABER data set is now over 13 years long, and measurements are still ongoing. Therefore, this data set offers the opportunity to investigate decadal variations of GW parameters in the middle atmosphere, such as the 11-year solar cycle. This multi-year variation is highlighted in Fig. 10 by displaying a 25-month running mean of GWSTA and absolute GWMF averaged over the latitude band 25°S to 25°N (blue curves). In Fig. 10, a relatively weak oscillation with a period of around 10 to 11 years can be identified. This variation is likely related to the 11-year solar cycle or another decadal variation of the whole earth system. The first "valley" of values that are below the median value of the time series (blue horizontal line in Fig. 10) occurs in the years 2003/2004, and the second "valley" in the years 2013/2014 (see the orange vertical lines in Fig. 10). Between these two "valleys", values are increased, i.e., higher than the median value.

Several studies investigated the relation between solar cycle and GW activity before, for example, Ern et al. (2011), Li et al. (2016) and Liu et al. (2017): Ern et al. (2011) indicated that there is an anti-correlation between the 10.7 cm solar flux and SABER GW amplitudes. Similarly, Liu et al. (2017) found a negative response of SABER GW potential energy to solar activity at lower and middle latitudes. In another study, Li et al. (2016) used 14 years of collected radiosonde data, i.e., a different measurement technique. Also in this study an 11-year cycle of GW energy densities was found in the tropics around 15 to 20 km altitude. Like for the SABER GW observations, an anti-correlation with the 11-year solar cycle was observed, and it was suggested that the variation of GW energy densities might be related to an 11-year oscillation of convective activity. Nevertheless, given the shortness of the available data sets, long-term variations, such as decadal or 11-year solar cycle of GW activity are still uncertain.

## 4 Variations of GWSTA and absolute GWMF at high latitudes (50°N/S-80°N/S)

Due to the yaw cycle of the TIMED satellite the SABER sampling is discontinuous at latitudes poleward of 50° for both hemispheres. To overcome this limitation, we perform sinusoidal fits for GWSTA and absolute GWMF for all latitudes of the SABER data considering the four dominant periods. With this method, the latitude range is extended and a global picture of the distributions of the four dominant periods in GWSTA and absolute GWMF can be obtained. The sinusoidal fit equation is expressed as



$$
\begin{aligned}
y(t) \quad = \quad & A_{TAV}\sin(\frac{2\pi}{4}t) + B_{TAV}\cos(\frac{2\pi}{4}t) + A_{SAV}\sin(\frac{2\pi}{6}t) + B_{SAV}\cos(\frac{2\pi}{6}t) \\
& + A_{AV}\sin(\frac{2\pi}{12}t) + B_{AV}\cos(\frac{2\pi}{12}t) + A_{QBO}\sin(\frac{2\pi}{26}t) + B_{QBO}\sin(\frac{2\pi}{26}t) + \text{offset}
\end{aligned} \tag{3}
$$

In Equation (3), $y(t)$ defines the monthly-mean time series from February 2002 to January 2015. The time $t$ is assumed to be in units of months. At the latitudes 50°S to 50°N this time series has no gaps. However, at latitudes 55°S to 80°S the months of February, June, and October are missing from $y(t)$, whereas at latitudes 55°N to 80°N the months of April, August and December are missing from the time series. Fits are performed using a sine and a cosine part, the respective amplitudes are noted by $A_*$ and $B_*$. Spectral amplitudes as discussed for the FFT analysis are calculated as $\sqrt{A^2 + B^2}$. Indices of $* = TAV,\ SAV,\ AV,\ QBO$ denote the corresponding periods.

By using the approach of Eq. (3), the amplitudes of each of the four dominant periods of GWSTA and absolute GWMF can be determined in the whole SABER latitude range, including high-latitudes poleward of 55° (see Fig. 11). The patterns of the fit results are in good agreement with the FFT results in the latitude band from 50°S to 50°N that were previously shown (cf. Figs. 2, 3, 5, and 7). Furthermore, the patterns at high-latitudes (50° to 80° for both hemisphere) are consistent continues extensions to the range of 50°S to 50°N, which is covered continuously by SABER data.

Many dominant features that were already discussed in Sect. 3 are visible in Fig. 11. For example, the MSAO and MQBO signals of absolute GWMF in the equatorial mesosphere are found in Fig. 11f and 11h. In addition, the spread of gravity wave activity by poleward propagation at increasing altitude can now be followed to 70° latitude and 90 km. It is visible in the AV (Fig. 11g) and particularly well pronounced in the SAV and TAV. This also raises confidence in the enhanced values of absolute GWMF in the summer hemisphere poleward of about 60° latitude, between about 80 km and 90 km. However, the cold summer mesopause is particularly difficult for temperature retrievals, and a larger contribution (enhancement of the maximum) due to noise is expected (Ern et al., 2011, 2018).

## 5 Summary and conclusion

In this paper we spectrally analyze time series of the square root of GW square temperature amplitudes (GWSTA) and, for the first time, absolute values of GW momentum flux (GWMF) in order to identify the systematic inter-annual, annual, and intra-annual variations in the stratosphere and mesosphere. The monthly zonal mean data are obtained from observations of the SABER/TIMED experiment and cover 13 years (February 2002 to January 2015). The latitude range of 50°S to 50°N is continuously covered and is analyzed via FFT. The FFT results were validated for an exemplary altitude of 30km using a MEM analysis. Due to TIMED yaw maneuvers every ∼60 days, higher latitudes have gaps in the time series and are analyzed by sinusoidal fits. The results from both techniques are consistent.

Four main modes of temporal variability are identified: annual variation (AV), semiannual variation (SAV), terannual variation (TAV), and quasi-biennial oscillation (QBO). In addition, we find indications for a decadal variation that might be related to



the 11-year solar cycle. The main results and mechanisms causing the variations are discussed in the paper and are summarized in Fig. 12.

For the annual variation, the most obvious variation of GWSTA and absolute GWMF occurs at mid- to high-latitudes. Reason is a maximum of GWSTA and absolute GWMF that is related to the polar vortex in the respective winter hemisphere.

This annual variation is stronger in the southern hemisphere than in the northern hemisphere. Compared with high-latitudes, a weaker annual variation is located in the subtropics, related to a local maximum of GW activity caused by convective sources in the summer hemispheres. These GWs from the subtropics can undergo large poleward latitude shifts while the GWs propagate upward. Hence, the subtropical maximum of absolute GWMF obliquely shifts poleward with increasing altitude. The oblique propagation is likely an important factor for mesosphere dynamics: GWs taking this pathway circumvent the critical level

filtering between tropospheric westerlies and stratospheric easterlies in the summer mid and high latitudes. They have the potential to be a major factor in the wind reversal in the upper mesosphere. This is supported by the fact that at the height of the wind reversal, a phase reversal from lower-altitude winter maximum to higher-altitude summer maximum occurs.

For the semi-annual variation (SAV), the most prominent variation of GWSTA is located between 60 and 90 km altitude at mid- to high-latitudes in both hemispheres. The maxima of GWSTA occur in summer and in winter. The spectral feature

occurs in a region into which in winter GWs propagate upward in the favorable winds of the polar vortex (indicated by blue in Fig. 12a and 12b) and into which in summer GWs propagate obliquely from the subtropics (indicated by red). Thus a winter maximum of westward propagating GWs and a summer maximum of eastward propagating GWs overlap in this altitude and latitude region. As the waves propagate westward in winter and eastward in summer, this forms an annual variation of net GWMF, but shows up as an SAV in the absolute values of GWMF accessible from the observations.

Another noteworthy feature is an enhancement of SAV spectral amplitudes of GWSTA and absolute GWMF in the equatorial MLT region (see Fig. 12c). This variation is related to convective GWs and their selective filtering and dissipation in the stratosphere caused by the SAO in tropical winds (Ern et al., 2015).

The terannual variation (TAV) is rarely mentioned in previous studies; the only publications known to the authors are Krebsbach and Preusse (2007) and Shuai et al. (2014). Our study reveals that the TAV is not an independent mechanism, but is likely

due to the pulse-like occurrence of the subtropical convective GW sources. Since in these region a four-month long "active" phase of strong convection is followed by an eight month long relatively "calm" phase, spectral analysis finds enhanced harmonics of the annual cycle, in particular a strong TAV component. The TAV therefore emphasizes the convective sources with respect to broader maxima e.g. in the winter vortex. The occurrence of a TAV therefore highlights the poleward propagation of convective GWs from the subtropics up to 70° latitude around 90 km carrying large amounts of GWMF.

The QBO of GWSTA and absolute GWMF occurs in the middle stratosphere (30 to 45 km) at low latitudes (10°S–20°N). Maxima of GW activity are found in the years 2003, 2005, 2007, 2009, etc., which is closely linked to the easterly phase of QBO winds. The QBO signals are also found to extend toward mid-latitudes around 50° in the stratosphere of both hemispheres. Furthermore, a peak of enhanced QBO spectral amplitudes is found in the equatorial mesopause region. Detailed analysis indicates that the QBO in the low latitude mesosphere ("MQBO") is coupled with the QBO in the stratosphere ("SQBO"):





when GW activity in the stratosphere is strong, GW activity in the mesosphere is weak and vice versa. This means that there is an anti-correlation between the MQBO and the SQBO, similar as for the MQBO and SQBO of the zonal wind.

The systematic spectral analysis of SABER GWSTA and GWMF reveals interesting features caused by filtering and oblique propagation. It hence can be used as a stringent constraint how well such processes are reproduced by global models.

*Data availability.*  The data set of monthly zonal average gravity wave activity derived from SABER observations is available from the PANGAEA open-access world data center at https://doi.org/10.1594/PANGAEA.879658. SABER satellite data are also freely available from GATS Inc. at http://saber.gats-inc.com. The operational meteorological analyses used in our study were obtained from ECMWF (http://www.ecmwf.int).

*Author contributions.*  DC carried out the data processing, performed the analysis, drafted the manuscript and designed the figures. CS and
PP carried out the model simulations. ME provided SABER satellite data. All authors discussed the results and commented on the manuscript and figures.

*Competing interests.*  The authors declare that they have no conflict of interest.

*Acknowledgements.*  We thank the team of the SABER instrument for their continued effort in maintaining and continuously improving the SABER data set. Further, we would like to thank ECMWF for providing the operational analyses used in our study. SABER data were pro-
vided by GATS Inc. and are freely available at http://saber.gats-inc.com. Operational meteorological analyses were obtained from ECMWF (http://www.ecmwf.int). The work of D. Chen was supported by the National Natural Science Foundation of China (Grant No. 41305038, 41675039), and China Scholarship Council. The work of M. Ern was supported by the Deutsche Forschungsgemeinschaft (DFG, German Research Foundation) project PR 919/4-1 (MS-GWaves/SV), the work of Cornelia Strube is supported by PR 919/4-2 (MS-GWaves/SV), which are part of the DFG researchers group FOR 1898 (MS-GWaves). M. Ern was also supported by the DFG project ER 474/3-1 (TigerUC)
which is part of the DFG priority program SPP-1788 "Dynamic Earth".



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





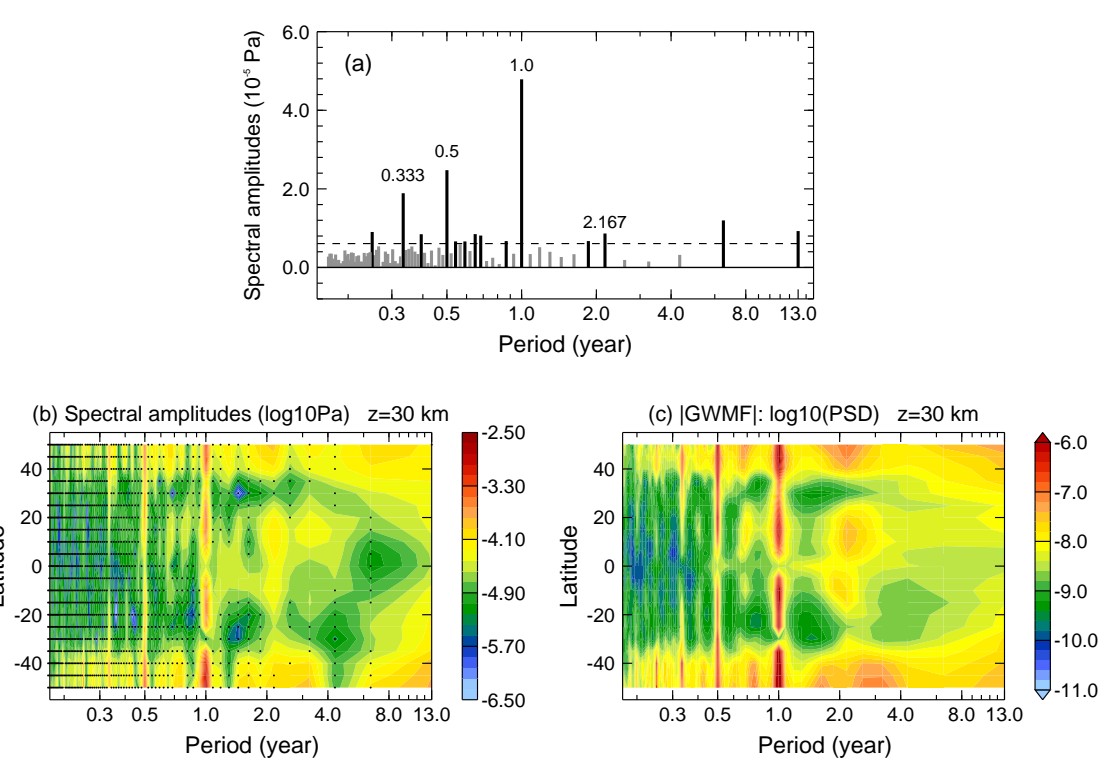

**Figure 1.** Example of an FFT spectrum of absolute GWMF performed at the latitude 45°N and altitude 50 km (a), and latitude dependence of the spectra at 30 km altitude for the latitude range 50°S–50°N (b). For comparison, also the MEM spectra at 30 km altitude are displayed for the latitude range 50°S–50°N (c). The significance level of 1.3 times the average spectral amplitude is shown as a horizontal dashed line in (a). Black dots in (b) indicate spectral amplitudes below the significance level of 1.3 times the average spectral amplitude.





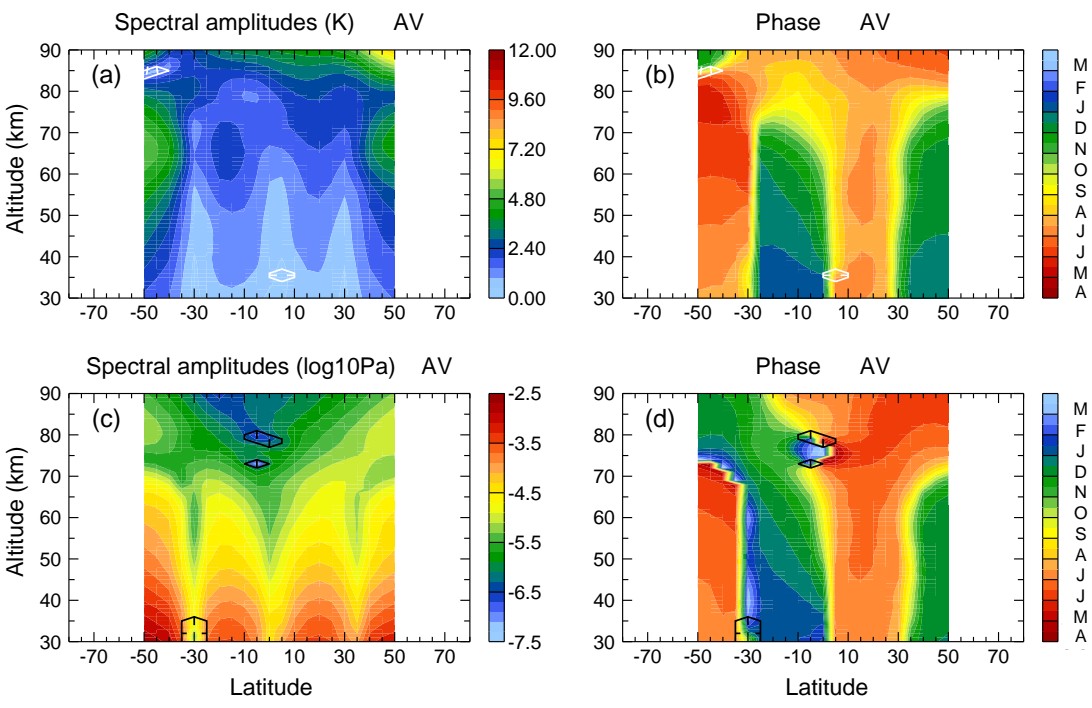

**Figure 2.** Altitude-latitude cross sections of spectral amplitudes (a, c, color code in K) and phase (b, d, color code in months) for annual variation of square-root of monthly zonal mean GW square temperature amplitudes and absolute GW momentum flux. Regions "downhill" of the white and black contour (in the direction of the ticks) are regarded as non-significant (i.e., values remain below 1.3 times the average spectral amplitude).





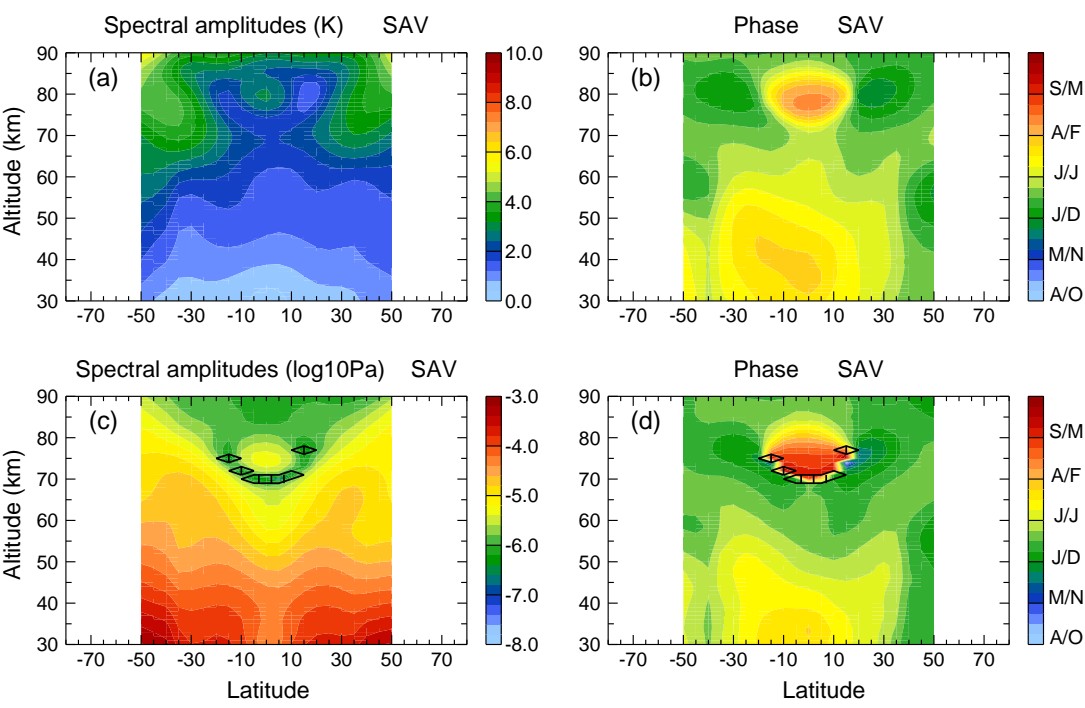

**Figure 3.** Same as Fig. 2, but for the semiannual variation.





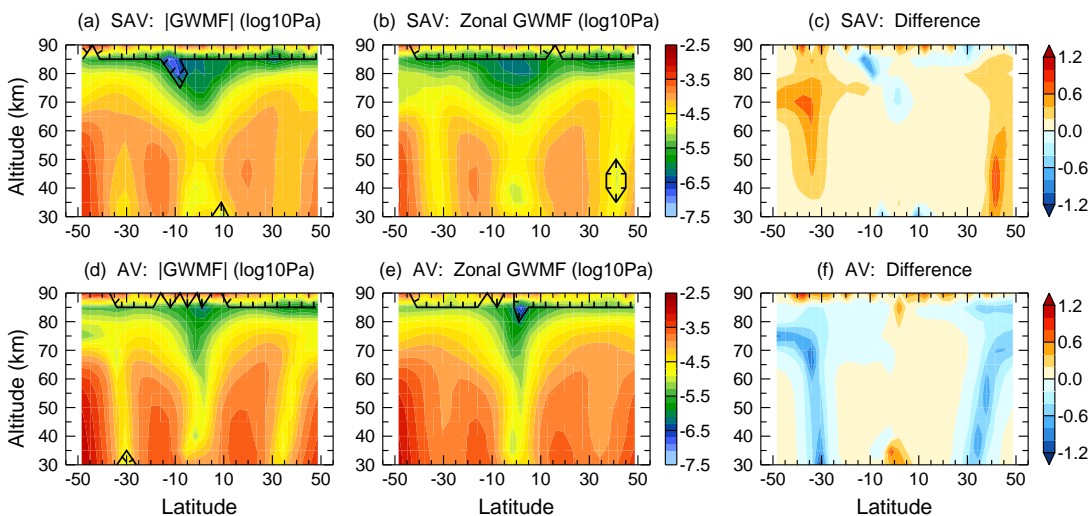

**Figure 4.** Spectral amplitudes of semiannual and annual variations for absolute GWMF, respectively (left column: a and d). The middle column (b and e) is the same as the left column, but for zonal GWMF (including direction). The differences of absolute GWMF and zonal GWMF spectral amplitudes for semiannual and annual variations, respectively, are shown in the right column (c and f).





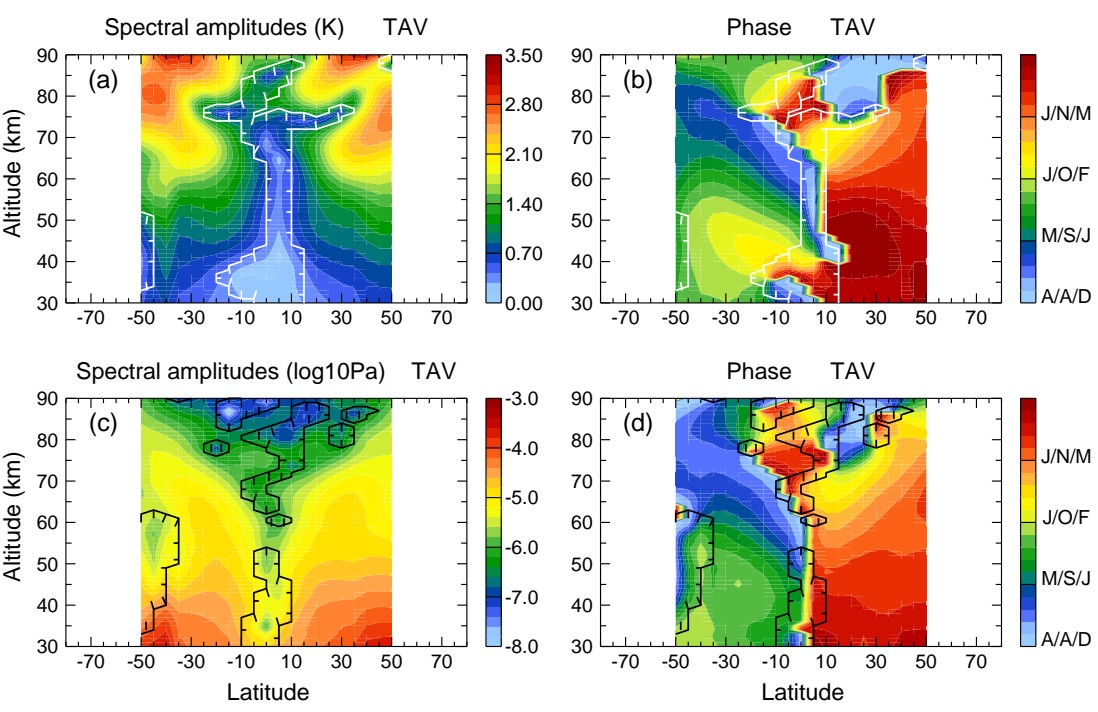

**Figure 5.** Same as Fig. 2, but for the terannual variation.





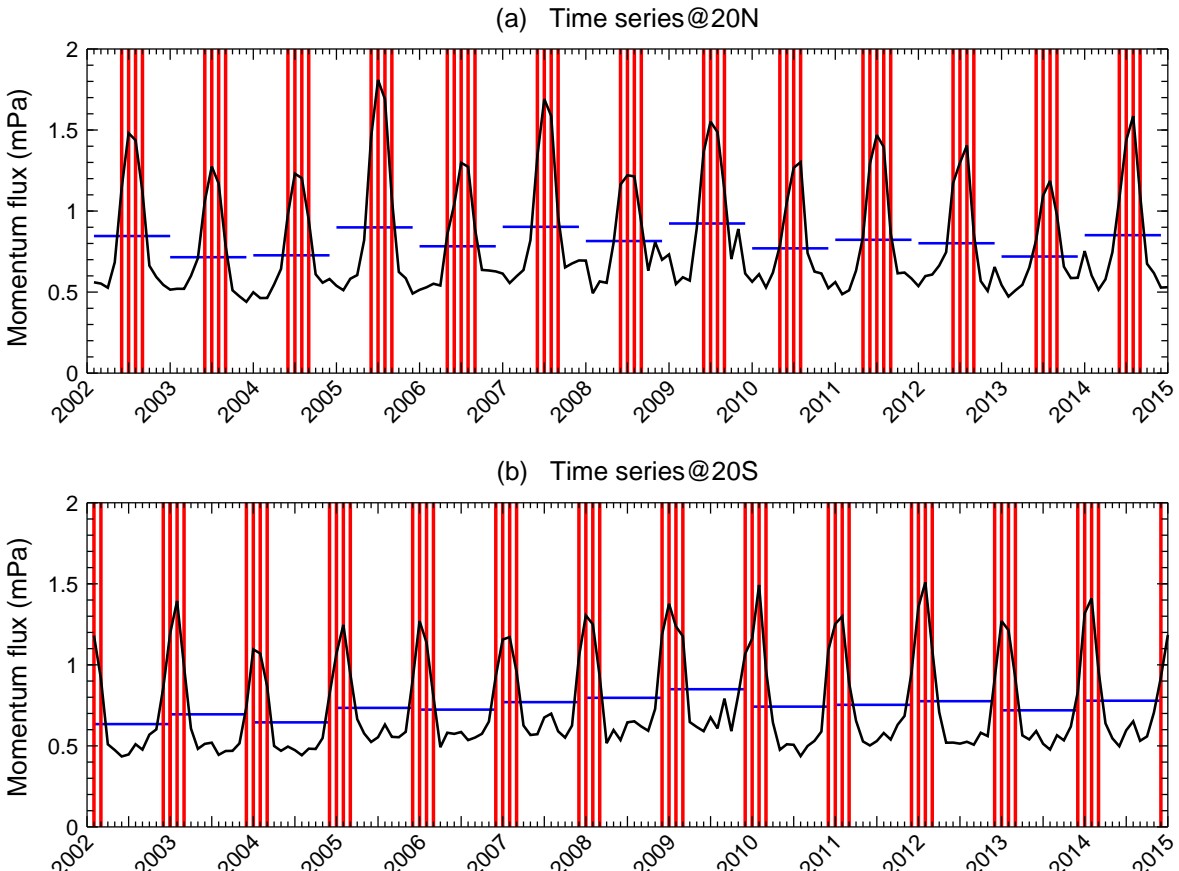

**Figure 6.** Time series of SABER absolute GWMF at 30 km for the latitudes 20°N (a) and 20°S (b). Blue horizontal lines represent the average values of each year. Red lines mark values that exceed the annual average values.





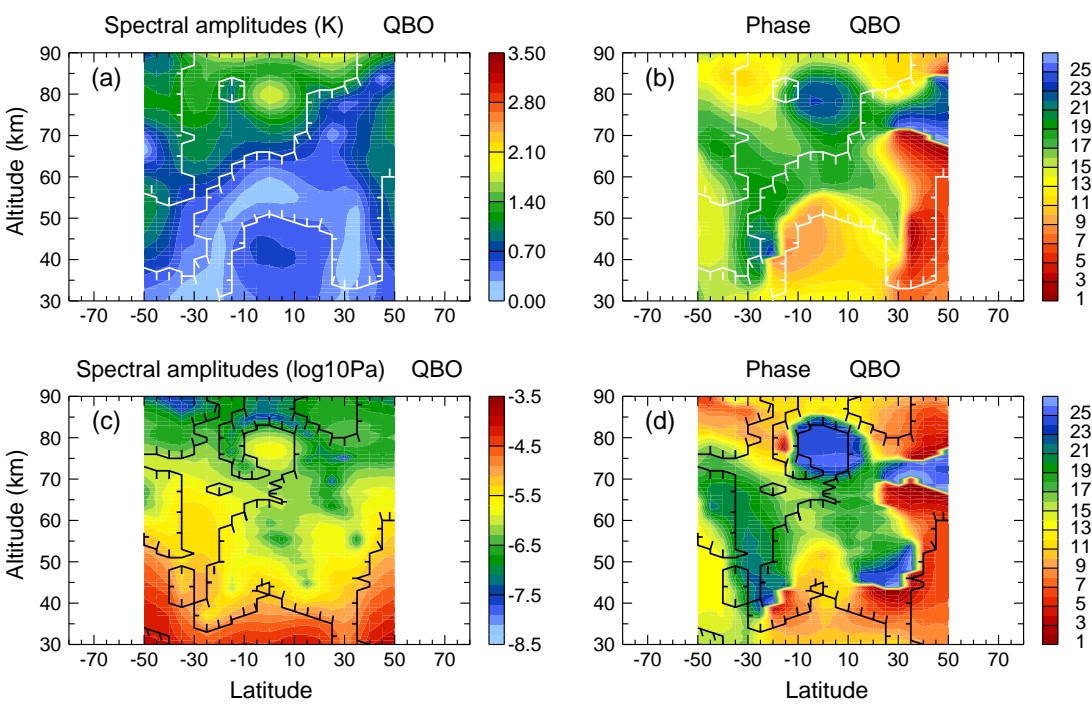

**Figure 7.** Same as Fig. 2, but for the QBO. The phase (b, d) color codes are in number index.



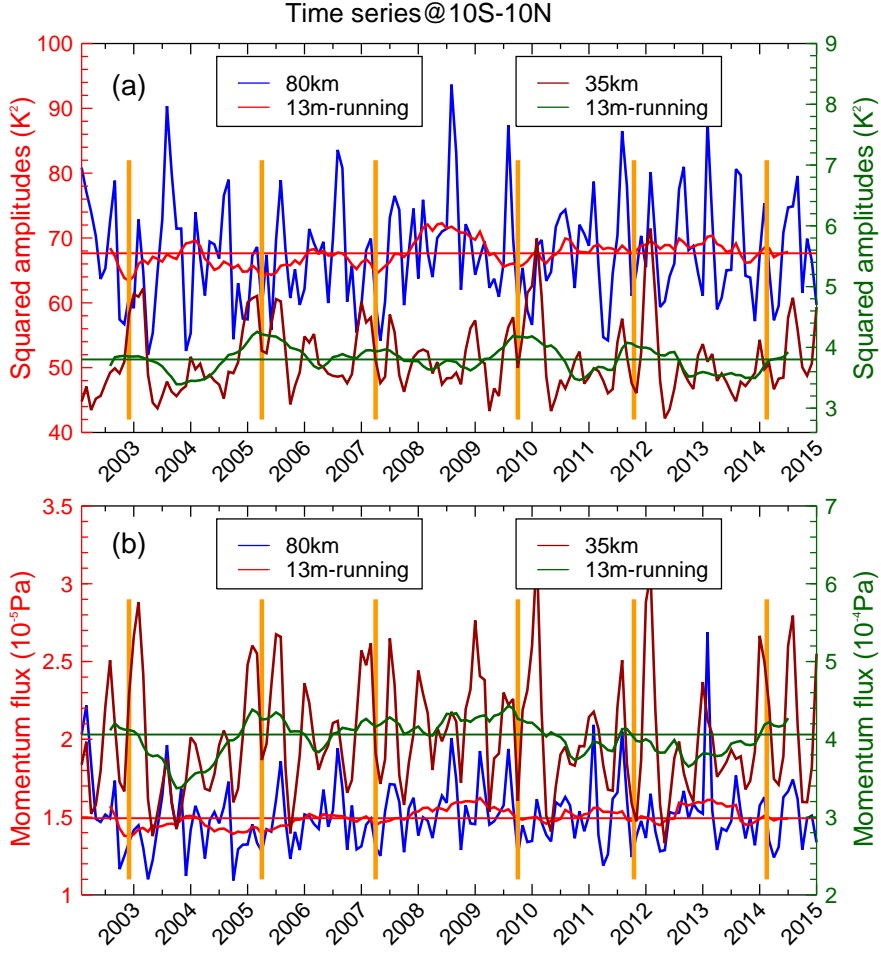

**Figure 8.** Time series of GW squared temperature amplitudes (a) and absolute momentum flux (b) at 80 km (blue lines) and 35 km (brown lines). Values are averaged over the latitude band $10°$S–$10°$N. 13-month running means (red lines for 80 km and green lines for 35 km, respectively) are given to highlight the overall QBO signals. The horizontal red and green lines are the median values. Maxima of GW activity at 35 km are marked by orange vertical lines in order to highlight the quasi-biennial variations.



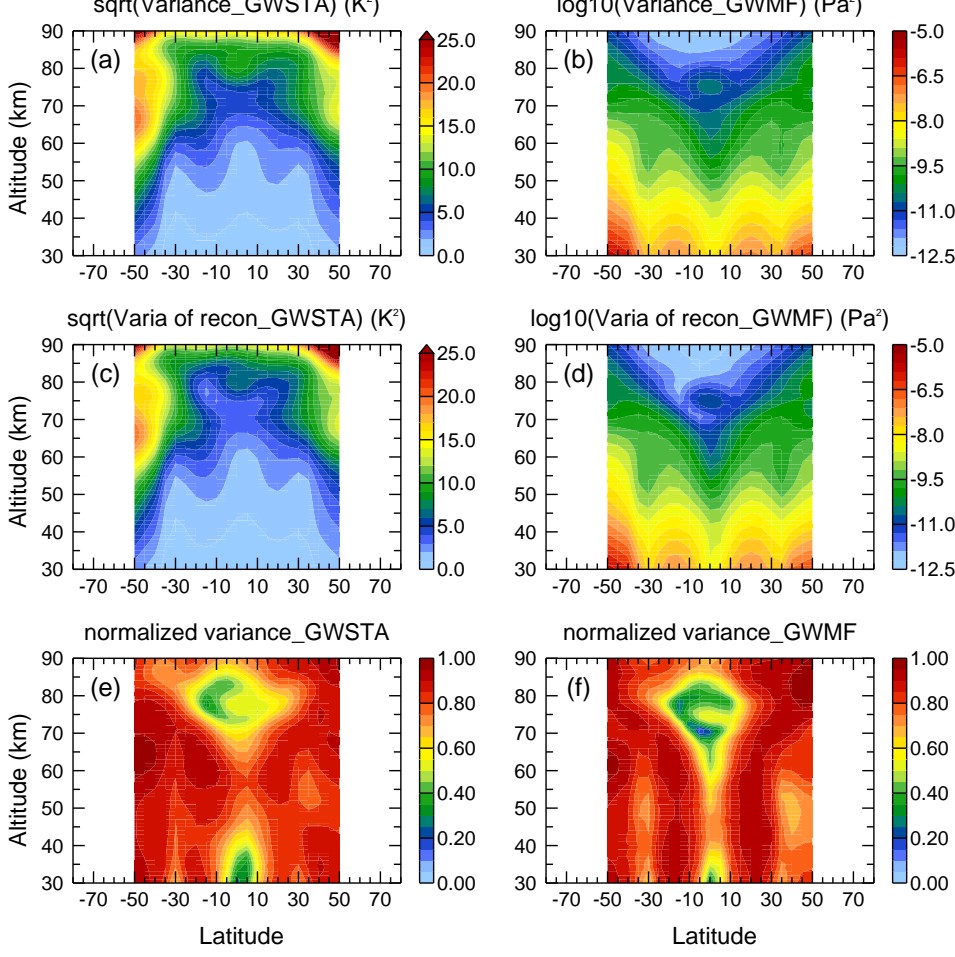

**Figure 9.** Altitude-latitude cross sections of the total variance contained in time series of SABER monthly-mean data for GW square temperature amplitudes (a) and absolute GW momentum flux (b), the variance of the time series reconstructed from the four main periods (c, d), and the fraction of variance explained by the four main periods (e, f).



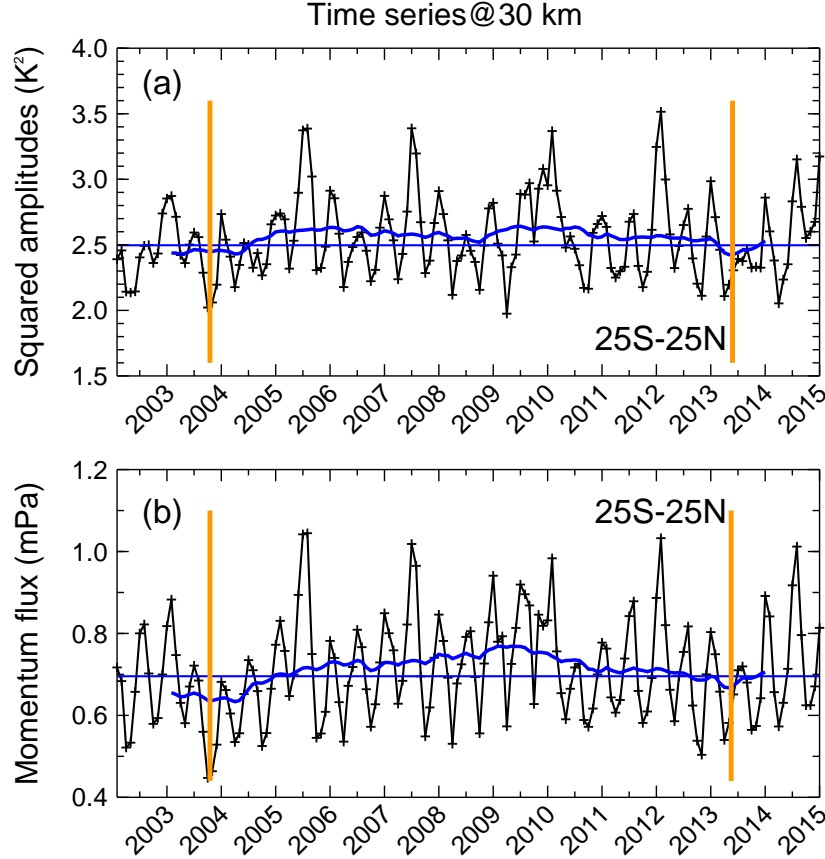

**Figure 10.** Time series of GW squared amplitudes (a) and absolute momentum flux (b) at 30 km. Values are averaged over the latitude band 25°S–25°N. Blue curves represent 25-month running means. Horizontal blue lines are the median values of the respective time series. Orange vertical lines mark the two minima of a potentially decadal variation.




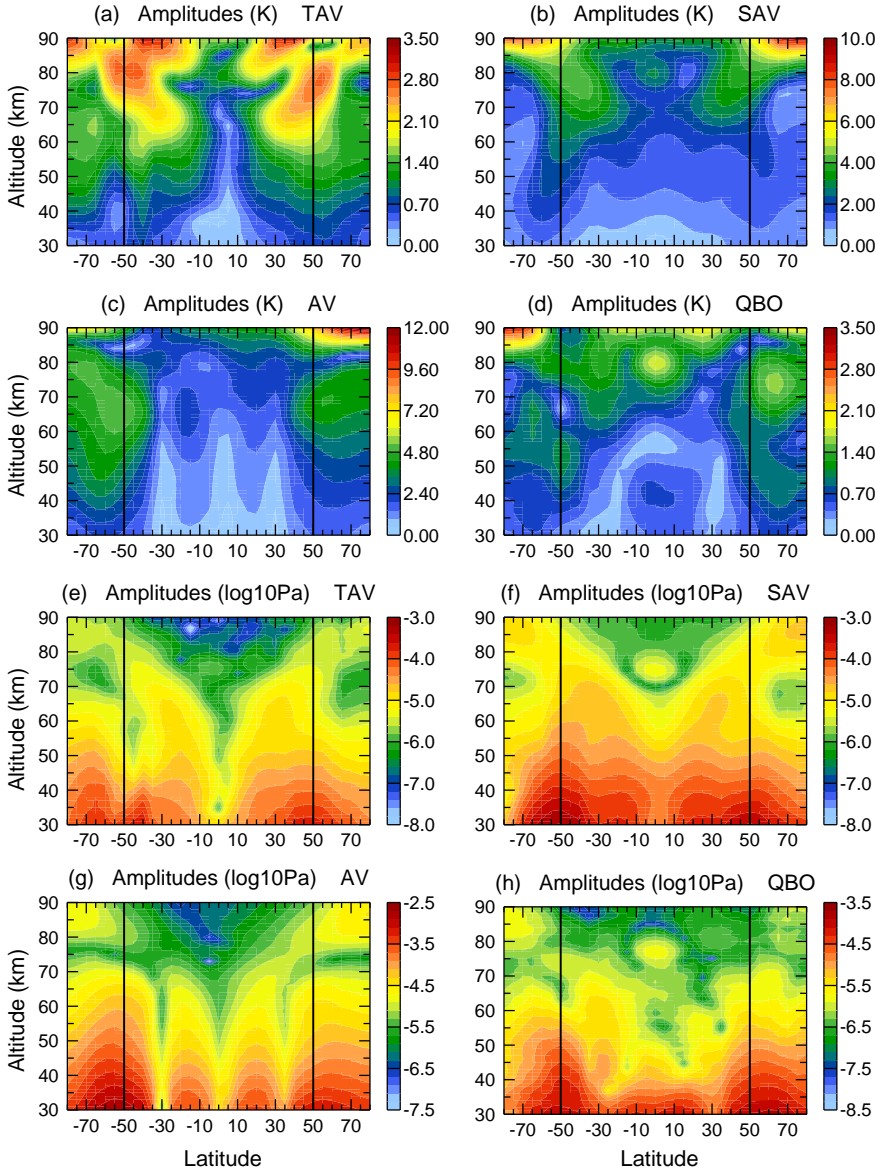

**Figure 11.** Altitude-latitude sections of spectral amplitudes for terannual, semiannual, annual, and QBO variation obtained by sinusoidal fits of the time series of square-root of monthly zonal mean GW square temperature amplitudes (a–d, unit in K) and GW absolute momentum flux (e-h, unit in log10(Pa)).



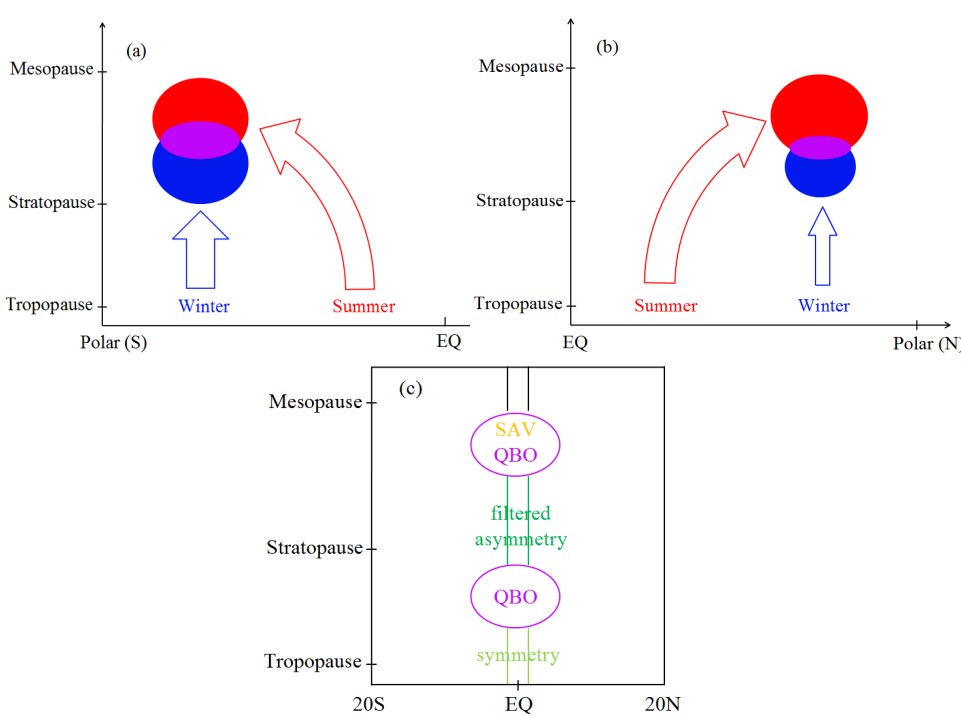

**Figure 12.** Schematic view for variations of GW activity on different time scales. Gravity waves that have their origin in an annual maximum of GW activity are indicated by colored thick arrows in (a) and (b). Blue arrows mean that during an annual variation GW activity is strongest in the winter season, while red arrows means stronger GW activity in summer. Many of these waves propagate obliquely. The purple shaded areas indicate the region where GWs of different season overlap and lead to variations other than an annual variation. For details, see text.