# Peer review of "Global analysis for periodic variations of gravity wave squared amplitudes and momentum fluxes in the middle atmosphere"

_Annales Geophysicae, 2019_

## Referee Comment (RC1) · Anonymous Referee #1 · 2 Apr 2019

In this study SABER satellite data are used to investigate the global distribution of spectral variability of monthly zonal mean gravity wave (GW) square temperature amplitudes (GWSTA) and absolute GW momentum flux (GWMF), i.e. no direction information. The knowledge about the global distribution of the temporal variations of GWMF and their sources is very valuable for models of all kinds. It cannot only help validate existing models but has great potential in improving GW-schemes and parametrizations. The authors focus here on four dominant frequencies: the annual, semiannual, terannual and on the QBO. The annual variation has two maxima in high and subtropical latitudes in each hemisphere resulting from filtering of GWs due to the strong winds in the polar vortex and from convective excitation of GWs in the subtropics. The semiannual variation has also 2 maxima at higher latitudes forming a crescent-shaped region with decreasing amplitudes towards the equator and also caused by convective GWs from the subtropical summer hemisphere propagating upward and poleward. However, since the GWMF has no direction information the semiannual variation is actually an annual variation which was proven by the authors applying the ray-tracer GROGRAT to a mixture of ERA-Interim and SABER data. The terannual variation has a peak around mid- to high latitudes in both hemispheres resulting from a combination of 4 consecutive month of a high occurrence rate of tropical convective systems followed by 8 consecutive "calm" months. The QBO variation has three maxima: The first one occurs in the equatorial stratosphere related to the well-known QBO winds. The second one appears at $50°$N/S in the stratosphere resulting from the modulation of GW filtering by the QBO modulated polar vortex and SSWs. The third one occurs in the equatorial mesosphere and results from the modulation of GWs by mesospheric QBO winds and may show that the mesospheric QBO is coupled to the stratospheric QBO. These four variations explain up to 80% of the total variance in most regions. Additionally a possible solar cycle effect on the GWSTA and GWMF temporal variations is discussed.

General comments:

This paper is very well written, the analyses methods are up to date and have been carried out with the utmost care. Only two aspects on why data are treated a special way should be clarified. Even though it would be nice to have a global GW momentum flux including propagation direction, the analysis method used here is the best we have at the moment as far as I know. The authors give also a very good and extensive discussion missing only a few points (see below) and summarize their results in clear schemes. Therefore I strongly recommend to publish this study after a minor revision.

Specific comments:

SABER data are available from January 2002 until today. Why do the authors use data

only between 2002 and 2015 and do not include at least also 2016 and 2017?

Monthly averaged GWSTA and GWMF have a grid resolution of $10°$ in longitude and $5°$ in latitude. To obtain values at each grid point the data are averaged in grid boxes of $30°$ in longitude and $20°$ in latitude. This means that the averaged grid boxes are three or four times larger than in the initial calculation. Is this really necessary, especially in the latitude direction?

Figure 1a shows the FFT spectrum of the GWMF at $45°$N. Besides the peaks discussed here (AV, SAV, TAV and QBO) there is also a peak at around 6 - 7 years which is even higher than that one of the QBO. I assume that this peak results from ENSO (El Nino southern oscillation). Please shortly discuss this peak even though you will not go into much detail in your further analyses.

Even though the Southern Hemisphere experienced only one SSW in the last decades namely in 2002, does make it a difference for the spectral analysis including or excluding this year? I am convinced of your results but I am also curious about the difference.

The described variance by the four variations in mid-latitude upper stratosphere is higher in the southern hemisphere than in the northern hemisphere. An additional cause for this effect might be a general much higher Rossby wave activity in the northern than in the southern hemisphere resulting in a much more variable polar vortex in the northern hemisphere and therefore in a much more variable GW filtering.

Figure 10 shows the GW variation related to the 11-year solar cycle. The authors calculated the median value of the time series and defined the beginning and ending of this solar cycle by the occurrence of "valleys". However, the dependency of the GW variation on the solar cycle is more pronounced in the GWMF than in the GWSTA since there is a third "valley" around 2009 even though it was still above the median. The solar minimum in 2009 was the lowest solar minimum of the last decades. Please discuss this the different behavior between GWMF and GWSTA around this solar minimum.

Technical corrections:

P2L24: . . .other sources are accounted for in a so-called . . .

P3L26: Li et al. (2016) further found an indication . . . (tense for consistency)

P11L20: . . . structures in Fig. 4a with Fig. 3c especially above 60km and Fig. . . . (similar structures are more obvious above 60km)

P12L21: . . . four month period (i.e. May/September/January) in the Southern Hemisphere. . . (for consistency with Fig. 5b)

P13L17: . . . use the number index 1 to 26 for the colorbar. -> Does month 1 presents January? Please clarify that in the text.

P14L20: One possible reason could be that the study of Krebsbach and Preusse (2017) . . . Figure 1: Example of an FFT . . .

Please also note the supplement to this comment:
https://www.ann-geophys-discuss.net/angeo-2019-31/angeo-2019-31-RC1-supplement.pdf

---

## Referee Comment (RC2) · Anonymous Referee #2 · 3 Apr 2019

Formal review of the manuscript

Global analysis for periodic variations of gravity wave squared amplitudes and momentum fluxes in the middle atmosphere

by Dan Chen et al.
* * *
The paper by Dan Chen et al. presents the temporal variation of square-root zonal average gravity wave squared temperature amplitudes (GWSTA) and the GW momentum flux (GWMF) based on 13 years of satellite data of the SABER instrument. A spectral

analysis is done focusing on the annual, semiannual, terannual oscillation up to quasi-biennial periods. The origin of these oscillations is interpreted in terms of different GW sources and the propagation pathways using a data set based on the combination of ECMWF data with GROGRAT ray-tracing. The paper is well written, the methods are adequately described and the results are properly discussed, therefore a publication in Annales Geophysicae is highly recommended. But nevertheless, some minor concerns should be addressed as indicated below before publishing.

Specific comments:

1) The authors focus on the annual, semiannual, terannual oscillation up to quasi-biennial periods as dominating oscillations. From Fig. 1a it can be seen that there are also oscillations with periods around 7 and 13 years visible. Please have a short discussion on that topic as these oscillations have even larger amplitudes than the QBO signal. Due to the length of the time series of 13 years, those peaks should be discussed in terms of the possible resolution within the FFT.

2) The difference of the variance based on the dominating periods between the northern and the southern hemisphere is related to SSW by the authors. It would be worth to discuss also the influence of planetary waves here as there are in general huge differences on both hemispheres due to the different land sea distribution.

Technical comments:

- Page 7, line 28: Spectral amplitudes in Fig. 1.b different compared to Fig. 1 a, for a reader it would be more intuitive to use the same axes as the color bar.

- Please improve the figure captions. Suggestion: Add labels GWSTA and GWMF for Fig. 2, 3, 5, 7, 11

- Page 35, Fig11: The colorbars in Figure 11 are very different. For a better comparison between each oscillation the color scale should be adjusted as equal as possible.

---

## Author Response (AR1)

Dear Dr Pisoft,

We would like to sincerely thank you very much for handling this manuscript. Please find below a point-by-point response to the reviews, a list of all relevant changes made in the manuscript, and a marked-up manuscript version.

Dan Chen

**A point-by-point response to the reviews:**

Anonymous Referee #1

**General comments:**
This paper is very well written, the analyses methods are up to date and have been carried out with the utmost care. Only two aspects on why data are treated a special way should be clarified. Even though it would be nice to have a global GW momentum flux including propagation direction, the analysis method used here is the best we have at the moment as far as I know. The authors give also a very good and extensive discussion missing only a few points (see below) and summarize their results in clear schemes. Therefore I strongly recommend to publish this study after a *minor revision*.

We thank the reviewer for the very positive recommendation and the detailed comments. We have taken all comments into account in the revised manuscript and give respective replies (in blue) below.

**Specific comments:**
1) SABER data are available from January 2002 until today. Why do the authors use data only between 2002 and 2015 and do not include at least also 2016 and 2017?

In our publication we use the referenced data set of the GRACILE climatology which is documented in Ern et al., ESSD, 2018 and which is publicly available under https://doi.org/10.1594/PANGAEA.879658, While SABER retrievals are continuously performed and released, the GW analysis is not a standard product and we here preferred to base on the published climatology. We have included a sentence in the revised manuscript with:

'This reference data set comprises the years 2002-2015 and we base our analysis on this period.' on page 5, line 12.

2) Monthly averaged GWSTA and GWMF have a grid resolution of 10° in longitude and 5° in latitude. To obtain values at each grid point the data are

averaged in grid boxes of 30° in longitude and 20° in latitude. This means that the averaged grid boxes are three or four times larger than in the initial calculation. Is this really necessary, especially in the latitude direction?

This again is due to our decision to use the GRACILE data set. For GRACILE the zonal means are calculated from the global maps. This has the advantage of a higher self-consistency of the data set. Furthermore, it harmonizes differences in the sampling. We have added an according sentence in the revised manuscript: 'In order to avoid biases through inhomogeneous sampling we base our zonal means on global maps.' on page 5, line 7.

3) Figure 1a shows the FFT spectrum of the GWMF at 45°N. Besides the peaks discussed here (AV, SAV, TAV and QBO) there is also a peak around 6-7 years which is even higher than that one of QBO. I assume that this peak results from ENSO (El Nino southern oscillation). Please shortly discuss this peak even though you will not go into much detail in your further analyses.

In Fig. 1a there is indeed a peak at around 6-7 years and also a peak at 13 years higher than QBO. From the horizontal axes in Fig.1a, it can be seen that the high resolution of periods analyzed by FFT concentrates below 3 years, while low resolution exists in 3-13 years. In addition, Fig. 1b shows the significant periods: in particular these are found for values below 3 years over a wide range of latitudes. For this reason, we only focus on the annual, semiannual, terannual, quasi-biennial periods as dominating oscillations.

Based on your suggestions, we have added a short discussion about this peak in our revised paper:
'In addition to the above-mentioned four major frequencies there are two peaks at around 6-7 years and 13 years. Due to the length of the time series of 13 years (156 months dataset), these are the two lowest frequencies (longest periods) which potentially could be resolved by FFT. These lowest frequencies are more likely to be influenced by leakage and aliasing. Therefore, we focus on the periods of 2.167, 1.0, 0.5 and 0.333 years, which are contained for several cycles and hence well constrained by our data. Potential explanations for the longer periods are ENSO which is an interannual oscillation of 2-7 years and the 11-year solar cycle.' on page 7, line28-33.

4) Even though the Southern Hemisphere experienced only one SSW in the last decades namely in 2002, does make it a difference for the spectral analysis including or excluding this year? I am convinced of your results but I am also curious about the difference.

Basing on your comment we have repeated the FFT excluding the year 2002. The resulting variance difference between excluding and including this year is shown in Fig. 1R of this reply. Figure 1Rc and 1Rd are the same as Fig. 9e and 9f in our manuscript and display the fraction of variance explained by the four main periods. The upper row shows the results excluding 2002, the middle row

the results including 2002 and the lower row the difference.

[Figure]

Figure 1R.    The variance of GWSTA and absolute GWMF for excluding (a, b) and including (c, d) the year 2002, and difference (e, f) between them.

Differences in the explained variance are generally small. In the latitude range of the southern hemisphere vortex there is a general increase of about 3% by excluding 2002. The largest difference is found in regions with a strong QBO signal, which is difficult to describe by a single, fixed period (cf. discussion in text) and which is better described when shortening the data set.

We have inserted a sentence in the revised manuscript:
'The exception from this is the split-event of the southern polar vortex in 2002. We have tested our analysis (not shown) and find only a very minor impact on our results. Different from this continuous descent of the phase of the southern polar vortex, the breakdown ...' on page 9, line 27-29.

5) The described variance by the four variations in mid-latitude upper stratosphere is higher in the southern hemisphere than in the northern hemisphere. An additional cause for this effect might be a general much higher Rossby wave activity in the northern than in the southern hemisphere resulting in a much more variable polar vortex in the northern hemisphere and therefore in a much more variable GW filtering.

Based on your suggestion and also based on comment 2 of the second reviewer we have added a short discussion about this effect in our revised paper:

'… is higher in the southern than in the northern hemisphere. Several effects may play a role. First, planetary wave activity is much higher in the northern than in the southern hemisphere. This results in higher variability in GW filtering. Likely more important for our time series of monthly zonal means is that this higher PW activity also frequently induces sudden stratospheric warmings, which terminate the ...' on page 15, line 16-19.

6) Figure 10 shows the GW variation related to the 11-year solar cycle. The authors calculated the median value of the time series and defined the beginning and ending of this solar cycle by the occurrence of "valleys". However, the dependency of the GW variation on the solar cycle is more pronounced in the GWMF than in the GWSTA since there is a third "valley" around 2009 even though it was still above the median. The solar minimum in 2009 was the lowest solar minimum of the last decades. Please discuss this the different behavior between GWMF and GWSTA around this solar minimum.

Since the coupling of the solar cycle is not really understood, it is difficult to speculate about any potential reasons. We have included a sentence to emphasize the difference between GWMF and GWSTA:

'This is more pronounced in absolute GWMF than GWSTA. The latter even has a small local minimum around 2009 which is close to the time of the solar minimum. The stronger solar cycle in absolute GWMF means that in particular GWs of short horizontal and long vertical wavelengths are modulated, which are more directly coupled to convective sources.' on page 15, line 29-32.

And, in addition:

'… convective activity consistent with the stronger modulation of absolute GWMF. The fact that we find some indications of absolute GWMF variations at timescales of around 11 years in the presence of other, stronger modulations, may be also due to 2008/2009 being the lowest solar minimum of the last decades. For more conclusive analyses several cycles of the investigated period would be needed which is still a long way to go from present day data sets.' on page 16, line 7-11.

**Technical comments:**
P2L24: …other sources are accounted for in a so-called…
P3L26: Li et al. (2016) further found an indication … *(tense for consistency)*

P11L20: …structures in Fig. 4a with Fig. 3c especially above 60km and Fig. …
*(similar structures are more obvious above 60km)*
P12L21: …four month period (i.e. May/September/January) in the Southern
Hemisphere… *(for consistency with Fig. 5b)*
Implemented as suggested.

P13L17: …use the number index 1 to 26 for the colorbar. -> *Does month 1
presents January? Please clarify that in the text.*
We have clarified this in our revised paper: 'According to the dominant QBO
period, we use 26 months for the phase display of the QBO. That means the
phase shifts roughly one month per calendar year with respect to e.g. January; in
other words, the fixed month allocation of phases we can use for the other
periods does not work for QBO and the month-index 1-26 indicates a different
calendar month in every year.' on page 13, line 28-31.

P14L20: One possible reason could be that the study of Krebsbach and Preusse
(2017) …
Figure 1: Example of an FFT …
Implemented as suggested.

Anonymous Referee #2

The paper by Dan Chen et al. presents the temporal variation of square-root
zonal average gravity wave squared temperature amplitudes (GWSTA) and the
GW momentum flux (GWMF) based on 13 years of satellite data of the SABER
instrument. A spectral analysis is done focusing on the annual, semiannual,
terannual oscillation up to quasi-biennial periods. The origin of these oscillations
is interpreted in terms of different GW sources and the propagation pathways
using a data set based on the combination of ECMWF data with GROGRAT
ray-tracing. The paper is well written, the methods are adequately described and
the results are properly discussed, therefore a publication in Annales
Geophysicae is highly recommended. But nevertheless, some minor concerns
should be addressed as indicated below before publishing.
We thank the reviewer for the very favorable recommendation and the
comments which helped to improve the manuscript. Please find the original
comment in black, the response in blue.

**Specific comments:**
1) The authors focus on the annual, semiannual, terannual oscillation up to
quasi-biennial periods as dominating oscillations. From Fig. 1a it can be seen
that there are also oscillation with periods around 7 and 13 years visible. Please
have a short discussion on that topic as these oscillations have even larger

amplitudes than the QBO signal. Due to the length of the time series of 13 years, those peaks should be discussed in terms of the possible resolution within the FFT.

Also reviewer 1 commented on the 7 and 13 year spectral content. In Fig. 1a there is indeed a peak at around 6-7 years and also a peak at 13 years higher than QBO. From the horizontal axes in Fig.1a, it can be seen that the high resolution of periods analyzed by FFT concentrates below 3 years, while low resolution exists in 3-13 years. In addition, Fig. 1b shows the significant periods: in particular these are found for values below 3 years over a wide range of latitudes. For this reason, we only focus on the annual, semiannual, terannual, quasi-biennial periods as dominating oscillations.

Based on your suggestions, we have added a short discussion about this peak in our revised paper:
'In addition to the above-mentioned four major frequencies there are two peaks at around 6-7 years and 13 years. Due to the length of the time series of 13 years (156 months dataset), these are the two lowest frequencies (longest periods) which potentially could be resolved by FFT. These lowest frequencies are more likely to be influenced by leakage and aliasing. Therefore, we focus on the periods of 2.167, 1.0, 0.5 and 0.333 years, which are contained for several cycles and hence well constrained by our data. Potential explanations for the longer periods are ENSO which is an interannual oscillation of 2-7 years and the 11-year solar cycle.' on page 7, line28-33.

2) The difference of the variance based on the dominating periods between the northern and the southern hemisphere is related to SSW by the authors. It would be worth to discuss also the influence of planetary waves here as there are in general huge differences on both hemispheres due to the different land sea distribution.

Also this point was raised by Reviewer 1 in her/his comment 5. Please note that we are here considering monthly zonal means which have already removed a part of the intermittency. Still, planetary waves may contribute and we have noted this in the revised manuscript.

Based on your suggestion and also based on comment 5 of the first reviewer we have added a short discussion about this effect in our revised paper:
'… is higher in the southern than in the northern hemisphere. Several effects may play a role. First, planetary wave activity is much higher in the northern than in the southern hemisphere. This results in higher variability in GW filtering. Likely more important for our time series of monthly zonal means is that this higher PW activity also frequently induces sudden stratospheric warmings, which terminate the ...' on page 15, line 16-19.

**Technical comments:**
-Page 7, line 28: Spectral amplitudes in Fig. 1b different compared to Fig. 1a, for a reader it would be more intuitive to use the axes as the color bar.
As we compare with the median, a linear axis seems more suited in Fig 1a. For the color plots we need a logarithmic axis to have sufficient contrast. In addition, the panels are for different altitudes. Therefore we will retain this figure as is.

-Please improve the figure captions. Suggestion: Add labels GWSTA and GWMF for Fig. 2, 3, 5, 7, 11.
As suggested, we have added the labels 'GWSTA' and 'GWMF' for Fig. 2, 3, 5, 7, 11 in our revised paper. Please see the modified figures in the revised paper.

-Page 35, Fig 11: The colorbars in Figure 11 are very different. For a better comparison between each oscillation the color scale should be adjusted as equal as possible.
According to your suggestion, we have adjusted the color bar. However, the spectral amplitudes of TAV and QBO are much smaller than those of AV and SAV. Therefore we now use the same color bar for TAV and QBO, and the other color bar for AV and SAV, respectively. Please see the revised manuscript for the adapted figures.

**A list of all relevant changes made in the manuscript:**

All of page and line numbers below are refer to the revised paper (without track changes).

(1) Page 2, line 24:    delete the word 'for'.

(2) Page 3, line 26:    find -> found.

(3) Page 5, line 7: added a sentence: 'In order to avoid biases through inhomogeneous sampling we base our zonal means on global maps.'

(4) Page 5, line 12: added a sentence: 'This reference data set comprises the years 2002-2015 and we base our analysis on this period.'

(5) Page 7, line 28-33: added a paragraph: 'In addition to the above-mentioned four major frequencies there are two peaks at around 6-7 years and 13 years. Due to the length of the time series of 13 years (156 months dataset), these are the two lowest frequencies (longest periods) which potentially could be resolved by FFT. These lowest frequencies are more likely to be influenced by leakage and aliasing. Therefore, we focus on the periods of 2.167, 1.0, 0.5 and 0.333 years, which are contained for several cycles and hence well constrained by our

data. Potential explanations for the longer periods are ENSO which is an interannual oscillation of 2-7 years and the 11-year solar cycle.'

(6) Page 9, line 27: added and revised some sentences: 'The exception from this is the split-event of the southern polar vortex in 2002. We have tested our analysis (not shown) and find only a very minor impact on our results. Different from this continuous descent of the phase of the southern polar vortex, the breakdown …'

(7) Page 11, line 30: added some words: '…especially above 60km…'

(8) Page 12, line 33: January /May/September -> May/September/January

(9) Page 13, line 28-31: revised and added some sentences: 'According to the dominant QBO period, we use 26 months for the phase display of the QBO. That means the phase shifts roughly one month per calendar year with respect to e.g. January; in other words, the fixed month allocation of phases we can use for the other periods does not work for QBO and the month-index 1-26 indicates a different calendar month in every year.'

(10) Page 14, line 34: added a word 'that'

(11) Page 15, line 16-19: revised and added some sentences: '… is higher in the southern than in the northern hemisphere. Several effects may play a role. First, planetary wave activity is much higher in the northern than in the southern hemisphere. This results in higher variability in GW filtering. Likely more important for our time series of monthly zonal means is that this higher PW activity also frequently induces sudden stratospheric warmings, which terminate the ...'

(12) Page 15, line 29-32: added some sentences: 'This is more pronounced in absolute GWMF than GWSTA. The latter even has a small local minimum around 2009 which is close to the time of the solar minimum. The stronger solar cycle in absolute GWMF means that in particular GWs of short horizontal and long vertical wavelengths are modulated, which are more directly coupled to convective sources.'

(13) Page 16, line 7-11: revised and added some sentences: '… convective activity consistent with the stronger modulation of absolute GWMF. The fact that we find some indications of absolute GWMF variations at timescales of around 11 years in the presence of other, stronger modulations, may be also due to 2008/2009 being the lowest solar minimum of the last decades. For more conclusive analyses several cycles of the investigated period would be needed which is still a long way to go from present day data sets.'

(14)
1. 'Example of an FFT…' -> 'Example of a FFT…'
2. The color scale of Fig. 1c was adjusted as equal as Fig. 1b.

5   (15)  added labels 'GWSTA' and 'GWMF' for figure 2.

(16)
1. Added labels 'GWSTA' and 'GWMF' for figure 3.
2. The color scale of Fig. 3a and 3c were adjusted as equal as Fig.2a and 2c,
10  respectively.

(17)  added labels 'GWSTA' and 'GWMF' for figure 5.

(18)
15  1. Added labels 'GWSTA' and 'GWMF' for figure 7.
2. The color scale of Fig. 7c was adjusted as equal as Fig. 5c.

(19)
1. Added labels 'GWSTA' and 'GWMF' for figure 11.
20  2. The color scale of Fig. 11b, 11f, and 11h were adjusted as equal as Fig. 11c,
11g, and 11e, respectively.

25  **From the next page, it is a make-up manuscript version.**

[revised manuscript text omitted]

In addition to the above-mentioned four major frequencies there are two peaks at around 6-7 years and 13 years. Due to 30 the length of the time series of 13 years (156 months dataset), these are the two lowest frequencies (longest periods) which potentially could be resolved by FFT. These lowest frequencies are more likely to be influenced by leakage and aliasing. Therefore, we focus on the periods of 2.167, 1.0, 0.5 and 0.333 years, which are contained for several cycles and hence well

constrained by our data. Potential explanations for the longer periods are ENSO which is an interannual oscillation of 2-7 years and the 11-year solar cycle.

[revised manuscript text omitted]
.  Several effects may play a role. First, planetary wave activity is much higher in the northern than in the southern hemisphere. This results in higher variability in GW filtering. Likely more important for our time series of monthly zonal means is that this higher planetary wave activity also frequently induces sudden stratospheric warmings, which terminate the

25  northern winter vortex at different times, but do not occur on the southern hemisphere (with the exception of the split-vortex event in 2002).

**3.6 Gravity wave variations potentially related to the 11-year solar cycle**

The SABER data set is now over 13 years long, and measurements are still ongoing. Therefore, this data set offers the opportunity to investigate decadal variations of GW parameters in the middle atmosphere, such as the 11-year solar cycle. This

30  multi-year variation is highlighted in Fig. 10 by displaying a 25-month running mean of GWSTA and absolute GWMF averaged over the latitude band 25°S to 25°N (blue curves). In Fig. 10, a relatively weak oscillation with a period of around 10 to 11 years can be identified. This variation is likely related to the 11-year solar cycle or another decadal variation of the whole earth system. The first "valley" of values that are below the median value of the time series (blue horizontal line in Fig. 10) occurs

in the years 2003/2004, and the second "valley" in the years 2013/2014 (see the orange vertical lines in Fig. 10). Between these two "valleys", values are increased, i.e., higher than the median value. This is more pronounced in absolute GWMF than GWSTA. The latter even has a small local minimum around 2009 which is close to the time of the solar minimum. The stronger solar cycle in absolute GWMF means that in particular GWs of short horizontal and long vertical wavelengths are modulated,

5  which are more directly coupled to convective sources.

Several studies investigated the relation between solar cycle and GW activity before, for example, Ern et al. (2011), Li et al. (2016) and Liu et al. (2017): Ern et al. (2011) indicated that there is an anti-correlation between the 10.7 cm solar flux and SABER GW amplitudes. Similarly, Liu et al. (2017) found a negative response of SABER GW potential energy to solar activity at lower and middle latitudes. In another study, Li et al. (2016) used 14 years of collected radiosonde data, i.e., a

10  different measurement technique. Also in this study an 11-year cycle of GW energy densities was found in the tropics around 15 to 20 km altitude. Like for the SABER GW observations, an anti-correlation with the 11-year solar cycle was observed, and it was suggested that the variation of GW energy densities might be related to an 11-year oscillation of convective activity . Nevertheless, given the shortness of the available data sets, long-term variations, such as decadal or 11-year solar cycle of GW activity are still uncertain consistent with the stronger modulation of absolute GWMF. The fact that we find some indications of

15  absolute GWMF variations at timescales of around 11 years in the presence of other, stronger modulations, may be also due to 2008/2009 being the lowest solar minimum of the last decades. For more conclusive analyses several cycles of the investigated period would be needed which is still a long way to go from present day data sets.

[revised manuscript text omitted]